# Guiding functions of the C-terminal domain of topoisomerase IIα advance mitotic chromosome assembly

Keishi Shintomi [1] & Tatsuya Hirano [1✉]

Topoisomerase II (topo II) is one of the six proteins essential for mitotic chromatid reconstitution in vitro. It is not fully understood, however, mechanistically how this enzyme regulates this process. In an attempt to further refine the reconstitution assay, we have found that chromosomal binding of *Xenopus laevis* topo IIα is sensitive to buffer conditions and depends on its C-terminal domain (CTD). Enzymological assays using circular DNA substrates supports the idea that topo IIα first resolves inter-chromatid entanglements to drive individualization and then generates intra-chromatid entanglements to promote thickening. Importantly, only the latter process requires the CTD. By using frog egg extracts, we also show that the CTD contributes to proper formation of nucleosome-depleted chromatids by competing with a linker histone for non-nucleosomal DNA. Our results demonstrate that topo IIα utilizes its CTD to deliver the enzymatic core to crowded environments created during mitotic chromatid assembly, thereby fine-tuning this process.

[1] Chromosome Dynamics Laboratory, RIKEN, Wako, Saitama, Japan. ✉email: hiranot@riken.jp

The mitotic chromosome, which consists of a pair of sister chromatids, is a gigantic molecular assembly that ensures accurate transmission of the duplicated genome into daughter cells. Since the first description of chromosome dynamics in the late nineteenth century, a myriad of efforts have been made to understand how mitotic chromosomes might be assembled at a mechanistic level[1,2]. One of the rational approaches to this longstanding question is biochemical identification and functional dissection of major chromosomal proteins. For example, pioneering studies had identified and characterized the chromosome scaffold, a remnant framework of mitotic chromosomes left after histone extraction[3]. It was shown later that the scaffold fraction contains topoisomerase II (topo II)[4] and ScII[5], the latter of which is currently known as a common subunit of condensins I and II[6]. Moreover, a series of experiments using cell-free extracts from *Xenopus laevis* eggs provided strong lines of evidence that topo IIα and condensins play essential roles in mitotic chromosome assembly[7–9]. Most importantly, chromatid-like structures can now be reconstituted in vitro by mixing a simple substrate with only six purified proteins including core histones, topo II, and condensin I along with three histone chaperones[10].

In recent years, alternative, genomics-based approaches have started shedding light on the three-dimensional organization of mitotic chromosomes. High-throughput chromosome conformation capture (Hi-C) techniques have demonstrated that the interphase chromatin landscape characterized by topologically associating domains (TADs) and compartments disappears upon mitotic entry, being replaced by mitosis-specific patterns composed of consecutive loops[11–15]. Together with mathematical modeling[15–17] and single-molecule experiments[18,19], it has been proposed that condensins make a linearly compressed array of loops through a mechanism termed loop extrusion, eventually resulting in the formation of rod-shaped chromosomes. Although such recent advancement has underscored the central importance of condensins in mitotic chromosome assembly, it is not yet fully understood how other chromosomal proteins like topo II collaborates with condensins to support this process. A comprehensive understanding of mitotic chromatid assembly will therefore require highly tractable experimental systems in which many parameters including protein concentrations, DNA topology, and ion atmospheres can be manipulated at will.

Here we report a systematic refinement of the mitotic chromatid reconstitution assay we described previously[10,20], which in turn allowed us to reveal hitherto-underappreciated functions of the C-terminal domain (CTD) of topo II. Furthermore, the powerful combination of the reconstitution assay, enzymological assays, and *Xenopus* egg cell-free extracts demonstrates that CTD-dependent intra-chromatid entanglement underlies chromatid thickening and that the CTD competes with a linker histone to prevent its abnormal actions on non-nucleosomal DNA.

## Results

**Refinement of the mitotic chromatid reconstitution assay.** In the mitotic chromatid reconstitution assay[10,20], *Xenopus* sperm nuclei were incubated with six purified proteins, and were directly transformed into a set of single chromatids without undergoing replication: the primary function of topo II in this reaction is thought to disentangle chromosomal DNAs intertwined with each other in the sperm nuclei and to individualize them for further structural changes. The final structures produced with the original protocol, in which budding yeast topo II was used, was a cluster of entangled thin chromatin fibers: the degree of chromatid individualization and thickening achieved in the reconstitution assay did not reach that observed in *Xenopus* egg cell-free extracts[20]. As an initial attempt to further improve the reconstitution assay, we first replaced budding yeast topo II with *X. laevis* topo IIα (hereafter referred to as topo IIα) in the current work. We confirmed that topo IIα could support chromatid reconstitution (Supplementary Fig. 1), but the morphology of the final products was no better than that of the products assembled using budding yeast topo II. Because it had been reported that topo II's actions are sensitive to buffer components in various in vitro assays[21–23], we then sought optimal buffer conditions for the reconstitution assay.

In the first set of assays, the concentration of $MgCl_2$ was titrated in a simplified buffer (Table 1). Sperm nuclei were incubated with a mixture of the six purified proteins supplemented with no $MgCl_2$ or increasing concentrations of $MgCl_2$. After a 150-min incubation, the resultant structures were fixed and analyzed by immunofluorescent microscopy. We found that the morphology of reconstituted chromatids was greatly improved as the $MgCl_2$ concentration in the reaction mixture increased (Fig. 1a): chromatid individualization greatly advanced and the disentangled chromatids substantially thickened at higher concentrations of $MgCl_2$. For quantitative comparison, fluorescent signal intensities were measured and the degree of their enrichment per unit of chromatin mass was estimated. Both topo IIα and condensin I was enriched on chromatin most efficiently in a buffer containing 5 mM $MgCl_2$ (Fig. 1b).

Secondly, we titrated the concentration of KCl in the reconstitution assay while keeping the $MgCl_2$ concentration at 5 mM. The results clearly demonstrated that, among all conditions tested, the reaction mixture containing 80 mM KCl produced the best-individualized set of chromatids (Fig. 1c). To verify the degree of individualization qualitatively, we defined and introduced an index (referred to as the individualization index), which was given by the quotient of the perimeter of a DAPI-positive area divided by the integral DAPI intensity in the same area (Fig. 1d). Notably, the highest enrichment of both topo IIα and condensin I on chromatid axes was observed at 80 mM KCl (Fig. 1c, e). At 200 mM KCl, chromatin binding of topo IIα was largely diminished and no fibrous chromatin structures were recognized.

**Table 1 Buffer compositions used for various chromatid reconstitution assays.**

|  | Supplementary Fig. 1 | Figure 1a | Figure 1b | Figure 1c | Figure 2 afterward |
|---|---|---|---|---|---|
| *X. laevis* topo IIα [nM] | 0–250 | 40 | 40 | 40 | 40 |
| $MgCl_2$ [mM] | 2.0 | 0–5.0 | 5.0 | 5.0 | 5.0 |
| KCl [mM][a] | 80 | 120 | 50–200 | 80 | 80 (150) |
| $KH_2PO_4$ [mM] | 40 | 0 | 0 | 0 | 0 |
| ATP [mM] | 2.0 | 2.0 | 2.0 | 0–1.0 | 2.0 |
| HEPES-KOH [mM], pH 7.7 | 20 | 20 | 20 | 20 | 20 |

[a]The KCl concentrations include their carryovers from storage buffers of protein preparations.

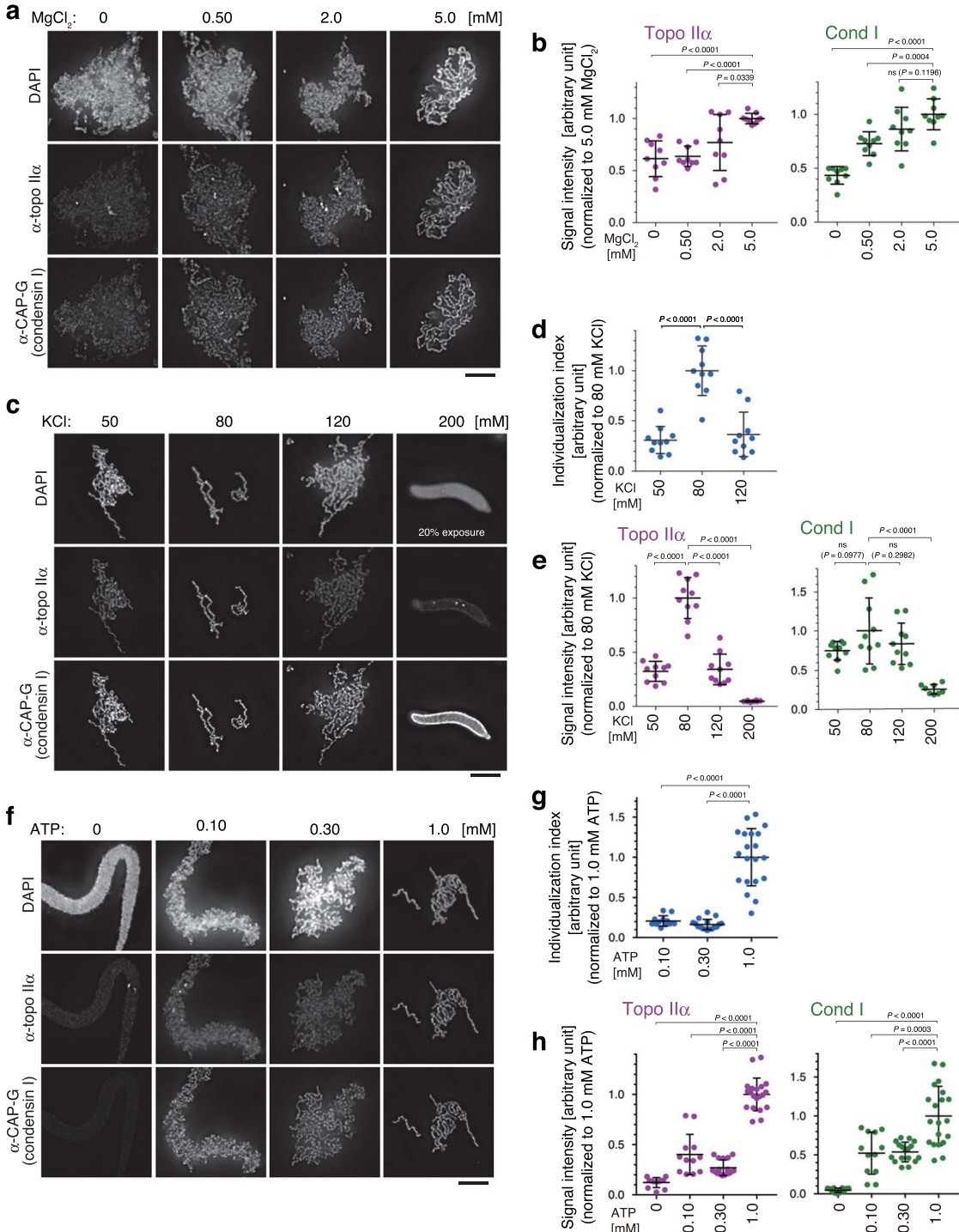

**Fig. 1 Manipulation of buffer conditions in the mitotic chromatid reconstitution assay. a**, **b** *Xenopus* sperm nuclei were mixed with a cocktail of six purified proteins (topo IIα, condensin I, a truncated version of histone H2A.XF-H2B dimer, Npm2, Nap1, and FACT) in a buffer containing no MgCl₂ or increasing concentrations of MgCl₂. The concentrations of other chemical ingredients in the buffer are listed in Table 1. After a 150-min incubation at 22 °C, the resultant structures were fixed and labeled with antibodies against topo IIα and CAP-G (a subunit of condensin I). DNA was counterstained with DAPI (**a**). Integral signal intensities of topo IIα or CAP-G were divided by those of DAPI in the same segmented region, and the values normalized to the average value in the reaction containing 5.0 mM MgCl₂ are plotted. The mean ± s.d. is shown (*n* = 9 clusters of chromatin). *P* values were assessed by two-tailed Welch's *t*-test (ns, not significant) (**b**). **c**–**e** KCl concentrations were titrated in the reconstitution assay. Immunolabeling and signal quantification were carried out as above (**c**, **e**). The perimeter of a DAPI-positive segmented area was divided by the integral intensity of DAPI in the same area, and the resultant values are plotted as individualization indices (**d**). The mean ± s.d. is shown. The sample sizes (*n*, the number of clusters of chromatin) are 10 (50, 80, and 120 mM KCl) and 8 (200 mM KCl). *P* values were assessed by two-tailed Welch's *t*-test (ns, not significant) (**d**, **e**). **f**–**h** ATP concentrations were titrated in the reconstitution assay (**f**). Individualization indices and signal intensities were analyzed as above. The mean ± s.d. is shown. The sample sizes (*n*, the number of clusters of chromatin) are 12 (0 and 0.10 mM ATP), 17 (0.30 mM ATP) and 20 (1.0 mM KCl). *P* values were assessed by a two-tailed Welch's *t*-test. ns not significant (**g**, **h**). Bars, 5 µm.

Lastly, we found that more than one millimolar of ATP was required for enrichment of topo IIα on chromatid axes as well as efficient chromatid individualization (Fig. 1f, g, h). Collectively, by narrowing down optimal buffer conditions, we successfully established a "second-generation" mitotic chromatid reconstitution assay that enables us to further dissect mitotic functions of topo IIα as shown below (Table 1).

**The CTD of topo IIα is required for chromatid thickening**. The titration experiments described above showed that chromatin binding of topo IIα is sensitive to buffer conditions. Because it had been reported that the CTD of topo IIα is important for localization on mitotic chromosomes in mammalian tissue culture cells[24], we decided to explore the functional contribution of this domain to mitotic chromatid assembly by using our reconstitution assay. To this end, we expressed and purified full-length (topo IIα-FL) and CTD-deleted (topo IIα-ΔCTD) versions of recombinant *X. laevis* topo IIα, in which two epitope tags (3 × FLAG and Strep II) were fused at their N- and C-termini (Fig. 2a, b).

We then introduced either topo IIα-FL or topo IIα-ΔCTD in the reconstitution assay and compared morphological changes of sperm nuclei between the two reactions over time (Fig. 2c). In both reactions, sperm nuclei rapidly swelled to form banana-shaped structures (10 min). Then entangled, thin chromatin fibers appeared and got individualized progressively (30 and 50 min). At 120 min when the reaction was completed, the individualization indices of the two reactions were indistinguishable from each other (Fig. 2d). We noticed, however, that the chromatids reconstituted with topo IIα-FL were thicker than those reconstituted with topo IIα-ΔCTD (Fig. 2e). We also found that, under the buffer condition used in the above experiment (80 mM KCl), the amount of topo IIα-ΔCTD detectable on the chromatids was far lower than that of topo IIα-FL (Fig. 2f, g; 80 mM KCl). In contrast, the same level of condensin I was detected between the two reactions. When the same set of assays was repeated in a buffer containing 150 mM KCl, topo IIα-FL still exhibited a partial activity to individualize chromatid fibers, but topo IIα-ΔCTD failed to do so, leaving banana-shaped structures that resemble those produced in a reaction containing no topo IIα at 80 mM KCl (Supplementary Fig. 1a). Note that both topo IIα-FL and topo IIα-ΔCTD were barely detectable on chromatin at 150 mM KCl (Fig. 2f, g; 150 mM KCl). Thus, topo IIα can promote chromatid individualization under relaxed buffer conditions and can do so even without its CTD at an optimal salt concentration. In contrast, chromatid thickening requires the CTD that facilitates chromosomal binding of topo IIα.

**The CTD of topo IIα is required for DNA catenation**. To gain further mechanistic insights into chromatid individualization and thickening, we compared the enzymatic activities of topo IIα-FL with those of topo IIα-ΔCTD. In the first setup (DNA decatenation assay), topo IIα-FL or topo IIα-ΔCTD was mixed with a catenated DNA substrate (kinetoplast DNA) at a low topo II/DNA ratio in the presence of ATP. In the reaction mixture containing 80 mM KCl, both enzymes generated decatenated circular DNAs via multiple rounds of the strand-passage reaction at a similar reaction rate. In contrast, when the KCl concentration was raised to 150 mM, topo IIα-ΔCTD failed to catalyze decatenation while topo IIα-FL is still proficient in this reaction (Fig. 3a, b). Because the conditions that support efficient DNA decatenation were comparable to those required for chromatid individualization in the reconstitution assay (Fig. 2c–g), it is reasonable to assume that topo IIα-catalyzed decatenation facilitates the chromatid individualization process.

In the second setup (DNA catenation/knotting assay), topo IIα was mixed with a nicked circular DNA substrate at a high topo II/DNA ratio and incubated for 10 min. AMP-PNP (instead of ATP) was then added, and the reaction mixtures were incubated for another 20 min to allow only a single round of the strand-passage reaction. At 80 mM KCl, topo IIα-FL generated two types of DNA products as shown previously[21,22,25]: fast-migrating knotted DNAs made from single DNA molecules and slowly-migrating catenated DNAs made from multiple DNA molecules (Fig. 3c, d). In contrast, topo IIα-ΔCTD produced only knotted DNAs under the same condition. At 150 mM KCl, both enzymes catalyzed neither knotting nor catenation. Thus, the conditions required for DNA catenation closely matched those for chromatid thickening in the reconstitution assay (Fig. 2c–g). We, therefore, hypothesized that CTD-mediated binding of topo IIα to chromatid axes might contribute to chromatid thickening by increasing the chance to generate entanglements between different chromatin loops within the same chromosomal DNA (hereafter, referred to as intra-chromatid entanglements).

If this hypothesis was correct, then topo IIα's enrichment on chromatid axes would no longer be essential once sufficient numbers of intra-chromatid entanglements had been generated to stabilize the resultant structures. Consistent with this view, our previous work had demonstrated that, after completion of chromatid assembly in *Xenopus* egg cell-free extracts, topo IIα can be readily displaced from chromatids without disrupting their overall morphology[7]. We applied a similar protocol to the chromatids assembled in the reconstitution assay. When the chromatids reconstituted with topo IIα-FL were exposed to 150 mM KCl, the signals of topo IIα on chromatid axes were largely diminished, but the overall morphology of the chromatids did not change as judged by DAPI stain and CAP-G labeling (Fig. 3e). When the same treatment was applied to the chromatids reconstituted with topo IIα-ΔCTD, their configuration was partly compromised and CAP-G localization became discontinuous. Thus, once the reconstitution reaction has been completed, topo IIα no longer has to stay on chromatid axes, suggesting that intra-chromatid entanglements, along with non-topo II protein components, are sufficient to maintain the rod-shaped chromatid structures.

**Depletion of topo IIα and nucleosomes produces sparklers**. We then asked whether the functional differences between topo IIα-FL and topo IIα-ΔCTD observed in the reconstitution assay could be reproduced in the *Xenopus* egg cell-free extracts. Depletion of endogenous topo IIα completely blocked chromatid individualization, leaving a banana-shaped chromatin mass derived from the *Xenopus* sperm nucleus (Supplementary Fig. 2a, b). Addition of recombinant topo IIα-FL into the depleted extract rescued such defects, forming a cluster of individualized and thick chromatids positive for topo IIα. In contrast, topo IIα-ΔCTD, despite being poorly detectable on chromatin, supported chromatid individualization, but not thickening. We also performed the same set of assays by using sperm nuclei isolated from the mouse *Mus musculus*[26]. This modified cell-free assay also demonstrated that the CTD is essential not only for topo IIα's enrichment on chromatids but also for chromatid thickening (Supplementary Fig. 2c).

The results described above prompted us to explore additional if any, layers of topo IIα's contribution to mitotic chromatid assembly under a more challenging condition that can be created only in the cell-free assay using mouse sperm nuclei. To this end, we depleted topo IIα and Asf1 from mitotic extracts (Δtopo IIα ΔAsf1; Supplementary Fig. 2a) and tested what would happen when topo IIα-mediated decatenation and nucleosome assembly

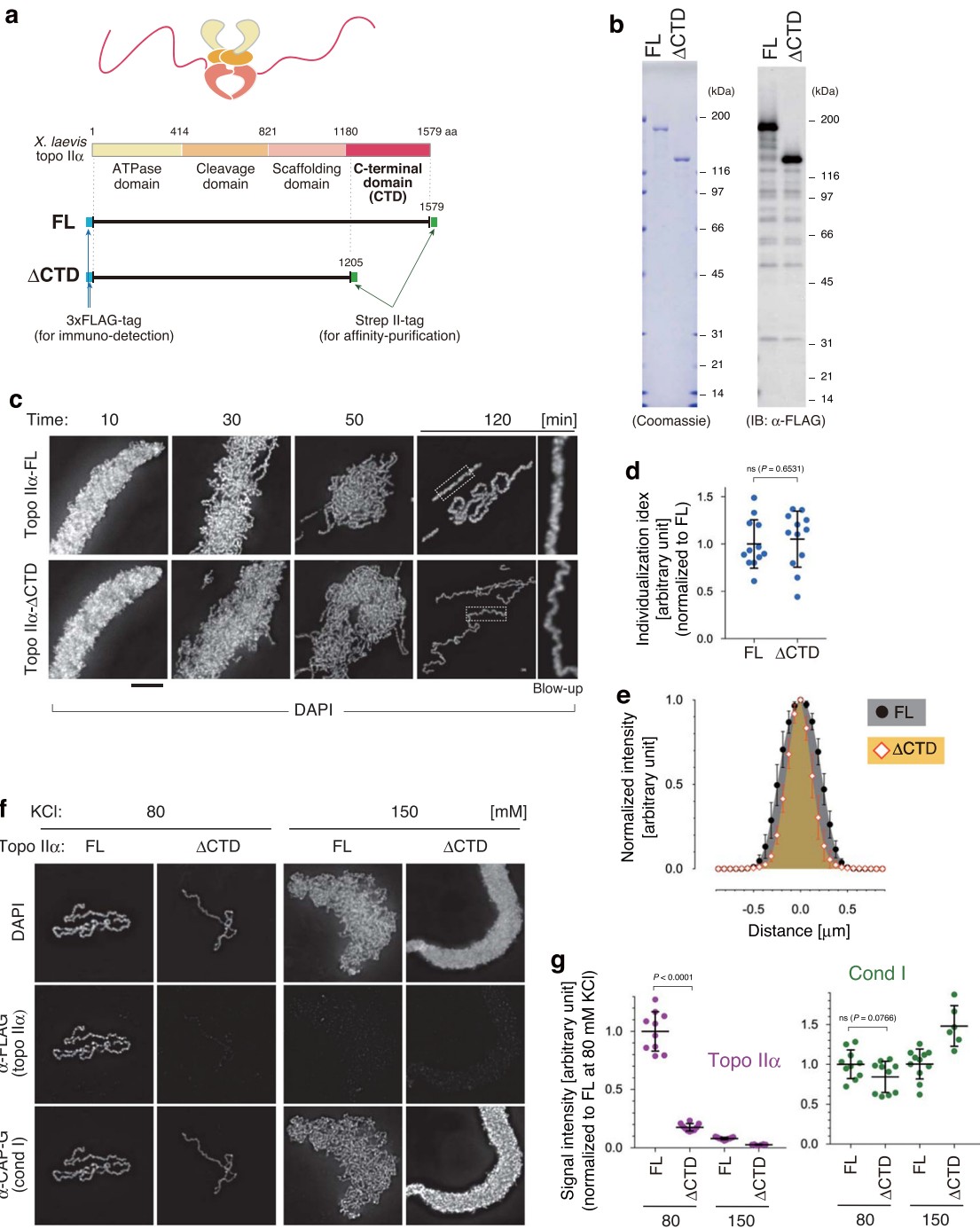

**Fig. 2 Topo IIα-ΔCTD is proficient in chromatid individualization but is deficient in chromatid thickening. a** Schematic presentation of structures of full-length (FL) and CTD-deleted (ΔCTD) versions of recombinant *Xenopus laevis* topo IIα. **b** Purified topo IIα-FL and topo IIα-ΔCTD were analyzed by SDS-PAGE and stained with Coomassie Blue. The same set of samples was also subjected to immunoblotting using anti-FLAG antibodies. This experiment was repeated three times with similar results. **c–e** *Xenopus* sperm nuclei were incubated in the reconstitution reaction mixture containing either topo IIα-FL or topo IIα-ΔCTD. At the indicated time points, the resultant chromatin was fixed and stained with DAPI. Blow-up images of cropped parts (indicated by the dashed rectangles in the original 120-min images) are shown on the right (**c**). Individualization indices at 120 min are plotted. The mean ± s.d. is shown ($n = 12$ clusters of chromatids). $P$ values were assessed by two-tailed Welch's $t$-test (ns not significant) (**d**). Profiles of normalized signal intensities of DAPI along lines drawn perpendicular to chromatid axes were analyzed. The mean ± s.d. is shown ($n = 15$ lines from 5 chromatids) (**e**). **f, g** Chromatid reconstitution assays were performed with topo IIα-FL or topo IIα-ΔCTD in a buffer containing 80 mM or 150 mM KCl. After a 150-min incubation, the resultant structures were fixed and processed for immunolabeling (**f**). Signal intensities of topo IIα and CAP-G on chromatids were analyzed. The mean ± s.d. is shown ($n = 9$ clusters of chromatin). $P$ values were assessed by two-tailed Welch's $t$-test (ns not significant) (**g**). Bars, 5 μm.

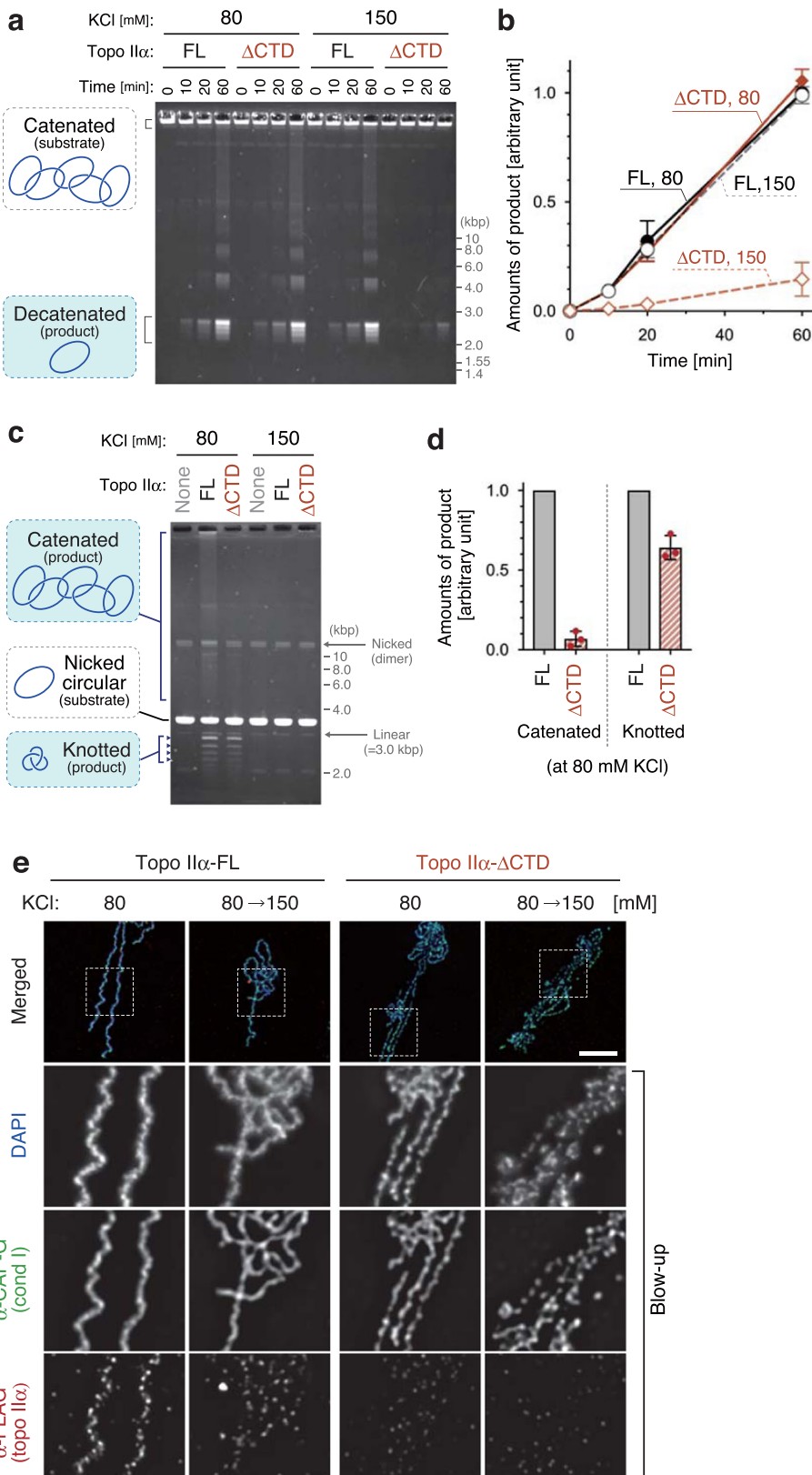

were simultaneously impaired. To our great surprise, we found that the Δtopo IIα ΔAsf1 extract produced a highly characteristic, compact structure that had not been observed or reported before: the majority of chromatin volumes was highly concentrated into a small central core, from which several protrusions radiated out (Fig. 4a, b). As expected, neither topo IIα nor histone H3 was

detected in this structure. Hereafter, we refer to this unusual structure as a sparkler because it resembled a type of small fireworks. Time-course analysis revealed that chromatin devoid of topo IIα and nucleosomes underwent a peculiar series of large-scale structural rearrangements at early time points (5–60 min), culminating in the formation of a sparkler at late time points

**Fig. 3 Topo IIα-ΔCTD is proficient in DNA decatenation but is deficient in DNA catenation. a, b** Catenated DNA (100 ng) was mixed with a low amount (40 ng) of topo IIα-FL or topo IIα-ΔCTD in a buffer containing 2 mM ATP, 5 mM MgCl$_2$, and either 80 mM or 150 mM KCl. After incubation at 22 °C, the resultant DNAs were recovered at the indicated time points, purified, and analyzed by agarose gel electrophoresis. The gel was stained with ethidium bromide (**a**). Intensities of decatenated DNA were quantified. The mean ± s.d. from three independent experiments is shown (**b**). **c, d** Nicked circular DNA (100 ng) was mixed with an excess amount (400 ng) of topo IIα-FL or topo IIα-ΔCTD in a buffer containing 5 mM MgCl$_2$, and either 80 mM or 150 mM KCl. After a 10-min incubation at 22 °C, the reactions were supplemented with 2 mM AMP-PNP and incubated for another 20 min. The DNAs were then purified and analyzed by agarose gel electrophoresis (**c**). Intensities of catenated and knotted DNAs were quantified. In each experiment, values of the topo IIα-ΔCTD reaction were normalized to those of the topo IIα-FL reaction. The mean ± s.d. from three independent experiments is shown (**d**). **e** Chromatid reconstitution assays were performed with topo IIα-FL or topo IIα-ΔCTD in a buffer containing 80 mM KCl. After a 150-min incubation at 22 °C, the mixtures were diluted by adding the same volume of reaction mixtures containing 80 mM KCl or 220 mM KCl so that the final concentrations of KCl became 80 mM or 150 mM. After another 20-min incubation at 22 °C, the resultant chromosomes were fixed and processed for immunolabeling. Blow-up images of cropped parts (indicated by the dashed rectangles in the merged images) are shown in grayscale. This experiment was repeated three times with similar results. Bar, 5 μm.

(90–150 min) (Fig. 4c). No fibrous structures were observed at any time point. Immunofluorescent labeling showed that condensins I and II displayed distinct distributions on the chromatin over time. In the final structure, dotty signals of condensin II distributed uniformly whereas strong signals of condensin I were detectable on the outer tip of protrusions (Fig. 4d).

**The CTD of topo IIα competes with the linker histone H1.8.** We then asked what protein(s) might be responsible for the formation of sparklers. The major protein constituents of mitotic chromatids assembled in control extracts include topo IIα, condensins, core histones, and H1.8 (also known as B4)[26]. H1.8 is an embryonic linker histone whose contributions to mitotic chromosome assembly had been tested in *Xenopus* egg cell-free extracts[27,28]. We reasoned that H1.8 might be the primary candidate that is responsible for sparkler formation because of the following two reasons. Firstly, neither condensin I nor condensin II was concentrated on the DAPI-dense core region within a sparkler (Fig. 4d). Secondly, it had been noticed previously that H1.8 transiently associated with mouse sperm-derived chromatin even before core histones were loaded, although this nucleosome-independent loading of H1.8 was diminished at late time points[26]. We, therefore, re-examined the localization of H1.8 on the products assembled under various conditions. As expected, H1.8 uniformly localized to the entire length of chromatids and chromatin fibers assembled in Δmock and Δtopo IIα extracts, respectively. Moreover, consistent with our previous study[26], H1.8 was not detectable on the nucleosome-depleted chromosomes assembled in a ΔAsf1 extract (Fig. 5a). Remarkably, however, we found that sparklers produced in a Δtopo IIα ΔAsf1 extract were positive for H1.8 despite the absence of core histones (Fig. 4a). H1.8 was detected on the DAPI-dense core and the inner tip of the protrusions of sparklers, whereas the major signals of condensin I were concentrated on their outer tip (Fig. 5b). These results suggest that the nucleosome-independent association of H1.8 with chromatin may play a key role in sparkler formation.

We then tested whether recombinant topo IIα-FL is indeed able to rescue the loss of endogenous topo IIα in the background of Asf1 depletion. When topo IIα-FL was added back into a Δtopo IIα ΔAsf1 extract, we observed a cluster of discrete, yet poorly compacted chromatids that was virtually identical to those assembled in a ΔAsf1 extract (Fig. 5c, d and Supplementary Fig. 3). As expected, topo IIα-FL and condensin I was enriched on these axes. In contrast, only a partial rescue was observed when topo IIα-ΔCTD was added back instead of topo IIα-FL: a core positive for DAPI and H1.8, a hallmark of the sparkler, was formed at the center of the structure, which was surrounded by condensin-positive chromatid axes. To further explore the

potential competition between topo IIα's action and sparkler formation, we set up a second complementation assay in which topo IIα-FL or topo IIα-ΔCTD was added back after sparkler formation was completed (Fig. 5e, f). Remarkably, topo IIα-FL was able not only to resolve pre-assembled sparklers but also to successfully support nucleosome-depleted chromatid assembly starting from such an unusually compacted substrate. Again, topo IIα-ΔCTD was less active in this protocol, failing to fully resolve pre-assembled sparklers. Our data demonstrate that the CTD of topo IIα competes with H1.8 to prevent sparkler formation as well as to dissolve pre-assembled sparklers.

**Discussion**
One of the primary motivations to initiate the current study was to refine the original protocol for the mitotic chromatid reconstitution assay using purified proteins[10,20]. In addition to replacing yeast topo II with *Xenopus* topo IIα, we surveyed for optimal buffer conditions as summarized in Table 1. The resultant mitotic chromatids reconstituted in this "second-generation" assay are morphologically indistinguishable, at the light microscopic level, from those assembled in crude egg extracts. Hence, the current data strengthen our original proposal that the fundamental process of mitotic chromatid assembly can be recapitulated with only six protein factors used here[10]. Our reconstitution assay is unique in the sense that it allows us to directly test the critical roles of surrounding ion atmospheres during the course of mitotic chromatid assembly. This contrasts with numerous previous studies that addressed this problem by measuring or manipulating ion atmospheres after chromosome assembly is completed[29–32]. Our results clearly demonstrate that chromosome binding of topo IIα, but not of condensin I, is highly sensitive to KCl concentrations (Fig. 1c, e) and that such a property heavily depends on its CTD (Fig. 2f, g). These observations have prompted us to uncover the specialized, context-dependent functions of the CTD during mitotic chromatid assembly as discussed below. The potential power of the reconstitution assay should be applied, in the future, to further dissecting how other chromosomal proteins such as condensin I and histones might work together in this seemingly complex process.

It has been thought that the major mitotic function of topo IIα is to resolve inter-chromatid entanglements[7,33–35]. Our results from both chromatid reconstitution assay and *Xenopus* egg cell-free extracts support this view (Supplementary Figs. 1 and 2b, c). The current study also provides strong lines of evidence that topo IIα has a second function in mitotic chromatid assembly: it promotes chromatid thickening at the late stage of chromatid assembly by generating intra-chromatid entanglements (Fig. 2c, e). This idea supports and extends the previous proposal that intra-chromatid entanglements contribute to maintaining the physical stiffness of mitotic chromosomes[36]. Considering the dual

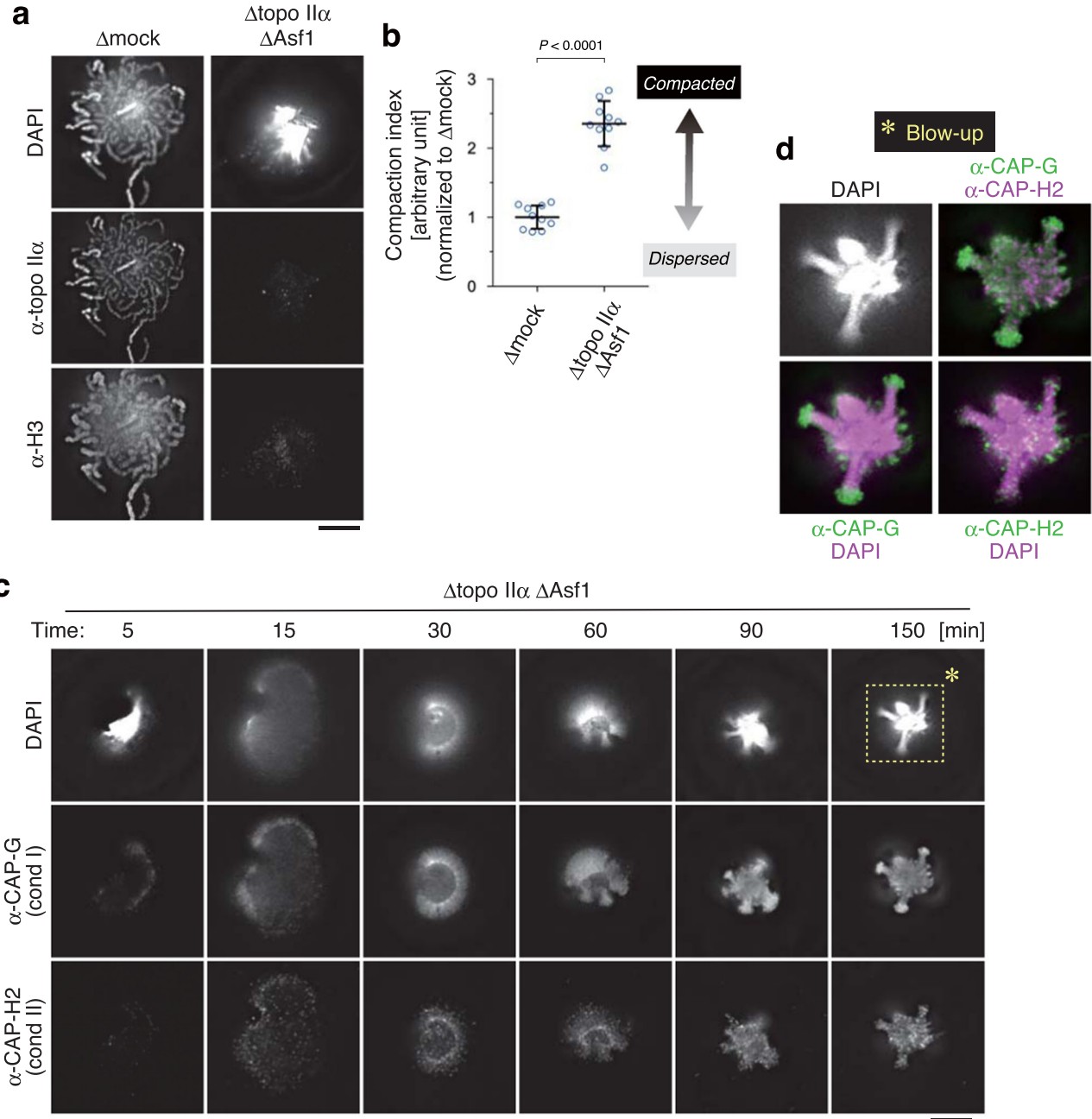

**Fig. 4 An unusual chromatin structure is produced in the cell-free extract depleted of both topo IIα and Asf1. a, b** Mouse sperm nuclei were incubated in a control extract (Δmock) or an extract depleted of both topo IIα and Asf1 (Δtopo IIα ΔAsf1) for 150 min and labeled with antibodies against topo IIα and histone H3. DNA was counterstained with DAPI (**a**). The compaction indices (the average DAPI intensities per unit area) are plotted. The mean ± s.d. is shown (*n* = 10 clusters of chromatin). *P* values were assessed by two-tailed Welch's *t*-test (**b**). **c, d** Mouse sperm nuclei were incubated in the Δtopo IIα ΔAsf1 extract at 22 °C. At the indicated time points, the reaction mixtures were fixed and labeled with antibodies against CAP-G (a condensin I subunit) and CAP-H2 (a condensin II subunit). DNA was counterstained with DAPI. This experiment was repeated three times with similar results (**c**). Blow-up images of the cropped part (indicated by the dashed rectangle in the original DAPI image at 150 min) are shown in grayscale (DAPI) and pseudo-colors (merged images for the indicated combinations) (**d**). Bars, 5 μm.

functions of topo IIα proposed here, the following mechanistic questions arise. Firstly, how does topo IIα choose the seemingly opposite reactions, namely, disentanglement and entanglement? The combination of the chromatid reconstitution and enzymological assays (Figs. 2 and 3) provides a simple answer to this question. On the one hand, topo IIα is able to support chromatid individualization even under the condition where it is barely detectable on the chromatids. This situation closely mimics the low enzyme/substrate ratio used in the decatenation assay. On the

other hand, the high enzyme/substrate ratio used in the catenation assay recapitulates the crowded environment where topo IIα is enriched along chromatid axes and supports chromatid thickening. It is reasonable to speculate that topo IIα needs a relatively long residence time to bring two DNA segments together and to catalyze strand-passage reactions under the latter, crowded condition. It should also be noted that topo IIα's action under this condition would be bidirectional, keeping an equilibrium between entanglement and disentanglement. The second

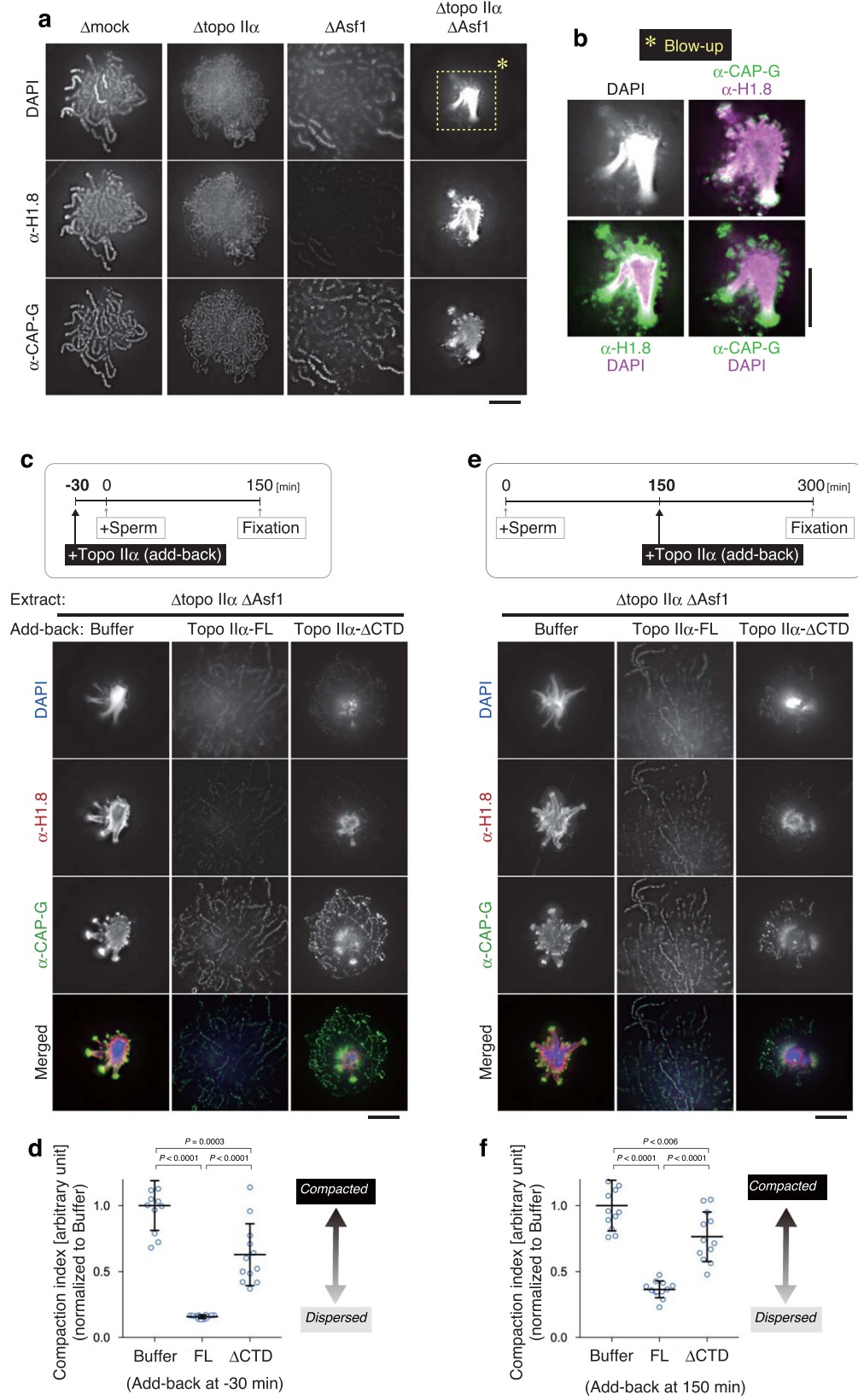

question is how topo IIα might collaborate with condensins during the chromatid thickening process. We envision that topo IIα allows entanglement between neighboring DNA loops created by condensins, thereby decreasing steric hindrance among DNA segments and facilitating further loop extrusion and chromatid thickening (Fig. 6a). It is also possible that intra-chromatid

entanglements are generated between DNA loops separated by a great genomic distance, for instance, when they are brought closer through a mechanism of helical winding of chromatid axes[15]. Our current results do not exclude the possibility that the CTD might contribute to chromatid thickening through a mechanism(s) other than intra-chromatid entanglement. Finally, it should be

**Fig. 5 Topo IIα-FL, but not topo IIα-ΔCTD, can efficiently rescue the defects observed in the extracts depleted of both topo IIα and Asf1. a, b** Mouse sperm nuclei were incubated in a control extract (Δmock) or an extract depleted of either topo IIα (Δtopo IIα), Asf1 (ΔAsf1), or both (Δtopo IIα ΔAsf1). After a 150-min incubation at 22 °C, the resultant structures were labeled with antibodies against the linker histone H1.8 and CAP-G. DNA was counterstained with DAPI. This experiment was repeated three times with similar results (**a**). Blow-up images of the cropped part (indicated by the dashed rectangle in the original DAPI image in a Δtopo IIα ΔAsf1 extract) are shown in grayscale (DAPI) and pseudo-colors (merged images for the indicated combinations) (**b**) **c, d** An extract depleted of topo IIα and Asf1 (Δtopo IIα ΔAsf1) was supplemented with either buffer, topo IIα-FL, or topo IIα-ΔCTD. After a 30-min incubation at 22 °C, mouse sperm nuclei were added to these extracts and incubated for another 150 min. The resultant chromatin structures were fixed and processed for immunolabeling with the antibodies indicated (**c**). The compaction indices were analyzed and are shown in Fig. 4b. The mean ± s.d. is shown ($n = 12$ clusters of chromatin). P values were assessed by two-tailed Welch's t-test (**d**). **e, f** Mouse sperm nuclei were first incubated in a Δtopo IIα ΔAsf1 extract to allow sparkler formation. At 150 min, the reaction mixtures were supplemented with either buffer, topo IIα-FL, or topo IIα-ΔCTD. After another 150-min incubation at 22 °C, the samples were fixed and processed for immunolabeling (**e**). The compaction indices were analyzed and are shown as above. The mean ± s.d. is shown ($n = 12$ clusters of chromatin). P values were assessed by two-tailed Welch's t-test (**f**). Bars, 5 µm.

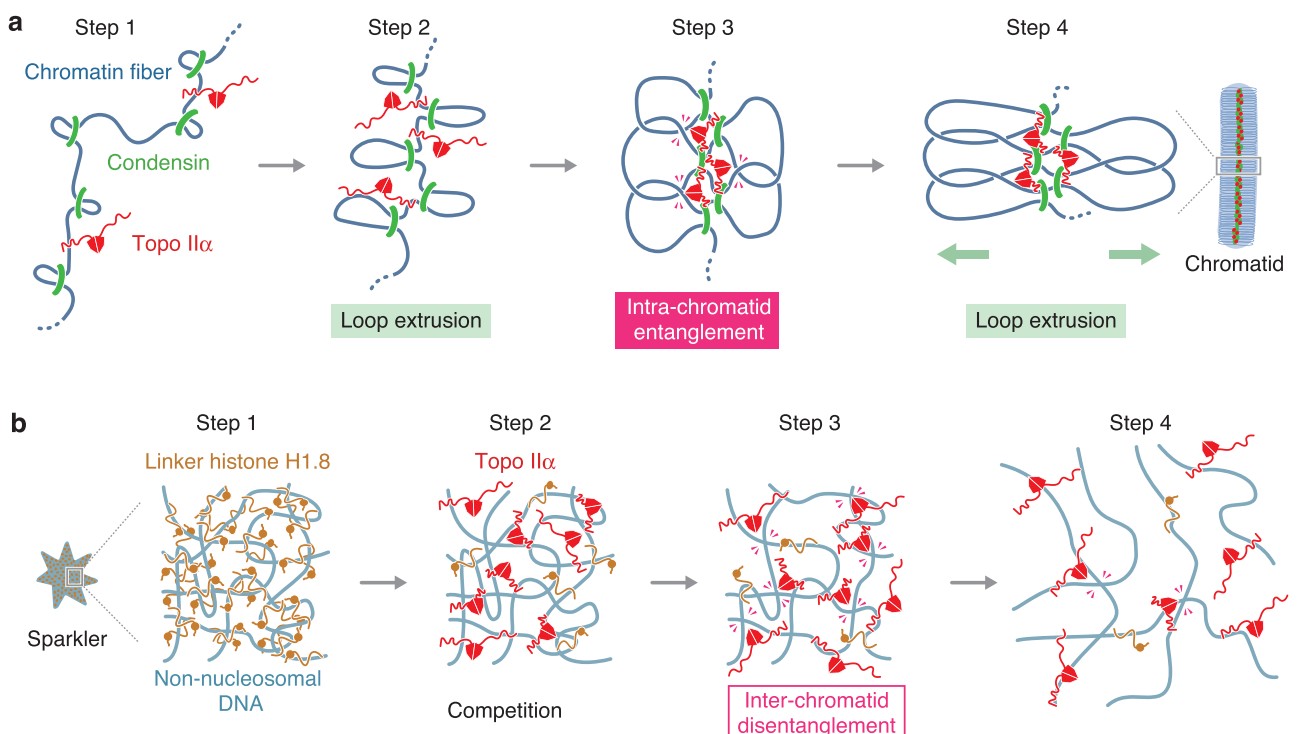

**Fig. 6 Models for the actions of topo IIα in crowded environments created during mitotic chromatid assembly. a** A model for chromatid thickening by collaborative actions of topo IIα and condensins. At the initial step of chromatid assembly, intra-chromatid entanglements are hardly generated because loops are distantly located from one another (step 1). As condensins extrude the loops, the distance between neighboring loops gets shortened (step 2), allowing chromatin-associated topo IIα to generate entanglements between loops (step 3: intra-chromatid entanglement). The resultant intra-chromatid entanglements in turn stabilize chromatid axes and may promote further extrusion of the loops and thickening of the chromatids (step 4). For simplicity, structural changes of an individualized, single chromatid are shown here. **b** A model for resolution of a pre-assembled sparkler. A sparkler is composed of multiple non-nucleosomal DNAs that are heavily entangled with each other and bound by the linker histone H1.8 (step 1). When topo IIα is added into the reaction, its CTD competes with the linker histone for the non-nucleosomal DNAs and delivers the enzymatic core into the inside of the sparkler (step 2). Topo IIα catalyzes disentanglement reactions there (step 3: inter-chromatid disentanglement), which in turn promotes further dissociation of the linker histone and resolves the sparkler (step 4). For simplicity, condensins' actions are not depicted here.

emphasized that our results are not consistent with a structural contribution of topo IIα to maintaining the architecture of mitotic chromosomes[35] because it can be experimentally displaced from the chromatids without affecting their structural integrity (Fig. 3e).

One of the most unexpected findings in the current study is that mitotic chromatin devoid of topo IIα and nucleosomes exhibits an unprecedented shape of structures, which we call sparklers. Our results reveal that the linker histone H1.8 is highly enriched in the central, DNA-dense region of sparklers (Fig. 5a, b). A previous single-molecule study demonstrated that the linker histone H1 compacts naked DNA at high stoichiometry[37] and

more recent studies have shown that the histone H1 phase separates to form biomolecular condensates in vitro[38,39]. Like other linker histone variants[40,41], H1.8 has a CTD that is predicted to be an intrinsically disordered region (IDR)[42]. It is, therefore, possible that H1.8, together with entangled non-nucleosomal DNAs, assembles sparklers through a mechanism of phase separation in the cell-free extracts. Consistent with this view, our results show that sparklers are dynamic and reversible structures: their assembly is prevented when topo IIα resolves inter-chromatid entanglements at early time points (Fig. 5c, d), and pre-assembled sparklers can readily be resolved by subsequent addition of topo IIα (Fig. 5e, f). Importantly, both actions

of topo IIα largely depend on its CTD. Taken all together, we suggest that the CTD of topo IIα, which itself is an IDR, competes with H1.8's IDR during the assembly and disassembly of sparklers (Fig. 6b). In this line, it is notable that early studies had reported preferential binding of topo II and the linker histone H1 to scaffold-associated regions (SARs), DNA elements predicted to be located at the base of chromatin loops[43,44]. Further studies will be required to understand the molecular mechanisms behind sparkler formation and possible involvements of other protein components including condensins I and II.

Our results presented here underscore specialized functions of topo IIα's CTD in crowded environments that are created during mitotic chromatid assembly (i.e., chromatid axes and sparklers). An emerging idea is that topo IIα utilizes its CTD to deliver its enzymatic core to proper loci on demand and to modulate its own action in different environments. This idea may have a broad implication in the evolution of IDR-mediated modulation of protein functions in general.

## Methods

### Preparation of recombinant topo IIα-FL and topo IIα-ΔCTD.
A cDNA encoding Xenopus laevis topo IIα was provided by Y. Azuma (University of Kansas, Lawrence, USA). DNA fragments encoding the full-length version (FL; amino acids 1–1579) and a CTD-deleted version (ΔCTD; amino acids 1–1205) of topo IIα were amplified by PCRs. PCR products were purified and cloned into Sma I-digested pXR504, a lab-made derivative of pFastBac1 (10360014, Thermo Fisher Scientific), using the Gibson Assembly system (E2611, New England Biolabs). The resulting plasmid vectors (termed pXR505 and pXR506) harbored the cDNA fragments encoding topo IIα-FL and topo IIα-ΔCTD, which were flanked by 3×FLAG-tag and Strep II-tag at their 5′- and 3′-ends, respectively. All the primers used for constructing these recombinant DNAs are listed in Supplementary Table 1. The Escherichia coli strain DH10EMBacY (Geneva Biotech) was transformed with either pXR505 or pXR506, and recombinant bacmid DNAs were produced. Baculoviruses were generated by introducing the bacmid DNAs into Sf9 cells (11496015, Thermo Fisher Scientific) and then amplified by another round of infection. Sf9 cells were then infected with the amplified viruses and grown in the SF900-III media (12658027, Thermo Fisher Scientific) at 27 °C for 72 h. The cells were harvested, resuspended in buffer ST (100 mM Tris-HCl [pH 8.0], 300 mM NaCl, 1 mM EDTA, 15% glycerol) supplemented with EDTA-free Complete Protease Inhibitor Cocktail (11873580001, Merck), and lysed by sonication. The lysate was clarified by centrifugation at 30,000g for 15 min and applied to a StrepTrap HP column (28907546, Cytiva). The column was washed with buffer ST, and bound proteins were eluted with buffer ST supplemented with 2.5 mM D-desthiobiotin (2-1000-005, Nacalai Tesque). For further purification of topo IIα-FL, the eluate was dialyzed against buffer SP-150 (20 mM Hepes-NaOH [pH 7.2], 150 mM NaCl, 10% glycerol), and loaded to a HiTrap SP HP column (17115101, Cytiva). The column was developed with a linear gradient of NaCl (150–600 mM). Peak fractions were collected and dialyzed against buffer KHG200/50 (20 mM Hepes-KOH [pH 7.7], 200 mM KCl, 50% glycerol). For purification of topo IIα-ΔCTD, the eluate from the StrepTrap column was dialyzed against Q-120 (20 mM Tris-HCl [pH 8.0], 120 mM NaCl, 10% glycerol), and loaded to a HiTrap Q HP column (17115301, Cytiva). The column was developed with a linear gradient of NaCl (120–500 mM). Peak fractions were collected and dialyzed against buffer KHG200/50. Small aliquots (20 μl) of the dialysates were snap-frozen in liquid nitrogen and stored at −80 °C until use. Once an aliquot was thawed, it was stored at −20 °C and used for up to 3 months. To evaluate the homogeneity of every preparation, an aliquot was analyzed by SDS-PAGE followed by Coomassie staining. A typical yield from 1 g insect cells was as follows: FL, 2.0 mg at a concentration of 6.0 μM; ΔCTD, 0.10 mg at a concentration of 2.0 μM.

### Mitotic chromatid reconstitution with purified proteins.
Mitotic chromatid reconstitution assays were carried out as reported previously[10,20] with some modifications. Briefly, the working concentrations of purified protein factors (other than topo IIα) used throughout the current study were as follows: Npm2, 60 μM; Nap1, 4.5 μM; dX-dB (a histone dimer of N-terminally deleted versions of the embryonic variants H2A.X-F and the canonical H2B), 900 nM; FACT, 360 nM; condensin I, 20 nM. The concentrations of topo IIα, salts and ATP were varied in some experiments and are summarized in Table 1. Note that the concentrations of HEPES and KCl in the reaction mixtures were calculated by considering those carried over from storage buffers of purified proteins. In all experiments, 10 mM phosphoenolpyruvate (P-0564, Sigma) and 100 units/ml pyruvate kinase (P-9136, Sigma) was added as an energy regeneration system along with the indicated concentrations of ATP. A reaction mixture was assembled by combining the abovementioned components and demembranated Xenopus sperm nuclei (1000 nuclei/μl) in a 20-μl volume and then incubated at 22 °C to allow chromatid reconstitution.

### Mitotic chromatin assembly in Xenopus egg extracts.
The high-speed supernatant of metaphase-arrested Xenopus egg extracts (M-HSS) was prepared as described previously[20]. Briefly, a low-speed supernatant was prepared by crushing unfertilized eggs in a buffer termed simplified XBE2 (20 mM Hepes-KOH [pH 7.7], 100 mM KCl, 2 mM MgCl₂, 5 mM EGTA) and further fractionated by centrifugation at 200,000g for 90 min. The resulting supernatant was collected and used as an M-HSS. Throughout the main text, the M-HSS was described as an egg extract. Demembranated Xenopus and mouse sperm nuclei were prepared as described previously[26,45]. Briefly, mature sperm heads of frogs were collected from testes and treated with lysophosphatidylcholine (L4129, Sigma). Mature sperm heads of mice were collected from cauda epididymis and treated with Streptolysin-O (S5265, Sigma). The resultant demembranated sperm nuclei were resuspended in buffer SMH (20 mM Hepes-KOH [pH 7.7], 2 mM MgCl₂, 250 mM sucrose) containing 0.4% BSA and 30% glycerol. In all assays, sperm nuclei were incubated with egg extracts at a final concentration of 1000 nuclei/μl at 22 °C. We found that structural changes of chromatin were almost completed by 120 min and that virtually no further discernible changes were observed afterward. The same was also true in all depletion or add-back assays. It is, therefore, reasonable to conclude that chromatin structure fixed after a 150-min incubation represents a terminal phenotype produced under a given condition. All animals were used in compliance with the institutional regulations and approved by the Animal Experimental Committee of the RIKEN Wako Institute.

### Antibodies.
Primary antibodies used in the current study were as follows: anti-topo IIα (in-house identifier: αC1-6, serum)[7]; anti-CAP-E (in-house identifier: AfR9-4, affinity-purified antibody)[8]; anti-CAP-G (in-house identifier: AfR11-3, affinity-purified antibody)[9]; anti-CAP-H2 (in-house identifier: AfR201-4, affinity-purified antibody)[46]; anti-Asf1(in-house identifier: R461-4, serum)[26]; anti-H1.8 (in-house identifier: #116-5, serum, provided by K. Ohsumi [Nagoya University, Japan])[47], anti-H3 (ab1791 [RRID: AB_302613], Abcam, for immunoblotting; CMA301 [RRID: AB_1977240], MBL, for immunofluorescence); anti-FLAG M2 (F-1804 [RRID: AB_262044], Sigma). Secondary antibodies used in the current study were as follows: Alexa Fluor 488 anti-mouse IgG (A11001[RRID: AB_2534069], Thermo Fisher Scientific); Alexa Fluor 568 anti-rabbit IgG (A11036 [RRID: AB_10563566], Thermo Fisher Scientific); horseradish peroxidase-conjugated anti-rabbit IgG (PI-1000 [RRID: AB_2336198], Vector Laboratories); horseradish peroxidase-conjugated anti-mouse IgG (PI-2000 [RRID: AB_2336177], Vector Laboratories).

### Immunodepletion.
An antibody was mixed with rProtein A-Sepharose beads (17127901, Cytiva) and incubated at room temperature on a rotating wheel for 60 min. The amount of each antibody coupled to 50 μl beads was as follows: for Asf1 depletion, 75 μl of anti-Asf1 serum; for topo IIα depletion, 75 μl of anti-topo IIα serum; for mock depletion, 100 μg of control rabbit IgG (I-5006, Sigma). The antibody-coupled beads were washed three times with KMH (100 mM KCl, 2.5 mM MgCl₂, 20 mM Hepes-KOH [pH 7.7]). Typically, a 100-μl egg extract was incubated with 50 μl of the antibody-coupled beads on ice for 30 min with occasional mixing (in some assays, the volumes of extracts and beads were changed proportionally). After two successive rounds of incubation, the supernatants were recovered by brief spin and used as depleted extracts. For double depletion of topo IIα and Asf1, four successive rounds of beads treatment (anti-topo II in the first and second rounds and anti-Asf1 in the third and fourth rounds) were performed. To evaluate the efficiency of depletion, an aliquot of each reaction was analyzed by immunoblotting.

### Immunofluorescence.
Chromatid assembly reactions were mixed with 10 volumes of 4% formaldehyde in KMH containing 0.1% Triton X-100 and incubated at 22 °C for 15 min. The resultant structures were centrifuged at 5000g for 10 min onto a coverslip through a 30%-glycerol cushion containing KMH using a custom-made fixed chromatid collection device[20]. For double labeling using a combination of primary antibodies derived from mice and rabbits, the coverslips were incubated with primary antibodies at a final concentration of 2.0 μg/ml (for affinity-purified one) or 1:2000 dilution (for serum) at 4 °C overnight, followed by incubation with fluorescently-labeled secondary antibodies at a final concentration of 2.0 μg/ml (labeled with Alexa Fluor 488 or 568) at RT for 2 h. In the case of double labeling using a pair of different rabbit antibodies, the coverslips were first incubated with an unlabeled primary antibody (anti-CAP-H2 or anti-H1.8), followed by Alexa568-conjugated anti-rabbit secondary antibody. The coverslips were then incubated with an excess amount (500 μg/ml) of non-immune rabbit IgG to quench the unreacted secondary antibody, and finally with anti-CAP-G that had been conjugated with Alexa488[48]. After counter-stained with DAPI, the coverslips were mounted on slides with VectaShield mounting medium (H-1000, Vector Laboratories), sealed with nail polish, and processed for fluorescence microscopy.

### Microscopy and image analyses.
Fluorescence microscopy was carried out using a DeltaVision microscope system (Cytiva), which consisted of an inverted fluorescence microscope (IX71, Olympus) equipped with a UPlanSApo 100×/1.40 oil-immersion lens and an sCMOS camera (pco.edge 4.2 m, PCO AG). The image stacks were acquired at a Z-step size of 0.2 μm and subjected to constrained iterative deconvolution with the SoftWoRx software (version 2.0.0, Cytiva). Deconvolved

image stacks were projected with three serial sections. The images were subjected to quantitative analyses by using the ImageJ software (version 2.0.0-rc-43/1.52n, https://imagej.nih.gov/ij). Estimation of the signal intensity was performed as follows: a chromatin-positive region was segmented in the DAPI channel by using the Threshold function; integrated density of DAPI and immune-fluorescence within the segmented region were measured by using the analyze particles function; and the latter value was divided by the former value for each region. Estimation of the other indices was performed in a similar way: a chromatin-positive region was segmented in the DAPI channel; for the individualization index, the perimeter of the region was divided by the integral intensity of DAPI; for the compaction index, the integral intensity of DAPI was divided by the area of the region. The line scan analysis was carried out as follows: to avoid biased measurements, we drew a first-line perpendicular to the chromatid axis at the widest region on a chromatid and then added two parallel lines on both sides of the first line at a 1-μm distance. DAPI densities along these three lines were scanned, and the same procedure was repeated for four additional chromatids ($n = 15$ from 5 chromatids).

**DNA decatenation assay**. One-hundred nanogram of a catenated DNA substrate (kinetoplast DNA; TG2013, TopoGEN) were mixed with 40 ng of recombinant topo IIα in a 10-μl volume (80 or 150 mM KCl, 5 mM MgCl$_2$, 20 mM HEPES-KOH [pH 7.7], and 2 mM ATP), and incubated at 22 °C. At the indicated time points, aliquots were taken and treated with SDS (0.5%) and proteinase K (1.0 mg/ml; P-4032, Sigma) at 37 °C for 1 h. The resultant DNAs were purified with phenol and separated by gel electrophoresis on a 0.8% agarose gel in TAE. After stained with ethidium bromide, fluorescent images were acquired by using an image analyzer (Amersham Imager 680 [version 2.0.0], Cytiva).

**DNA catenation/knotting assay**. For the preparation of a nicked circular DNA substrate, a 3.0-kb plasmid DNA (pBlueScript SK(-)) was treated with the nicking enzyme Nt.BspQI (R0644S, New England Biolabs). The nicked DNA (100 ng) was mixed with 400 ng of recombinant topo IIα in a 10-μl volume (80 or 150 mM KCl, 5 mM MgCl$_2$, and 20 mM HEPES-KOH [pH 7.7]) and incubated at 22 °C for 10 min. After supplemented with AMP-PNP (AppNHp; NU-407, Jena Bioscience) at a concentration of 2 mM, the reaction was incubated at 22 °C for another 20 min. The reaction was treated with SDS (0.5%) and proteinase K (1.0 mg/ml; P-4032, Sigma) at 37 °C for 1 h. The resultant DNAs were purified with phenol, separated by gel electrophoresis on a 0.8% agarose gel in TBE, and visualized with ethidium bromide. Fluorescent images were acquired as described above.

**Immunoblotting**. Protein samples were separated by SDS-PAGE and transferred onto a nitrocellulose membrane (Hybond-ECL, RPN303D, Cytiva) with submarine-type blotting equipment (Trans-Blot Cell, Bio-Rad). The protein-bound membrane was incubated with a 5% skimmed milk solution made in TBS-Tw (20 mM Tris-HCl [pH 7.5], 150 mM NaCl, 0.05% Tween-20) for 30 min and then probed with primary antibodies diluted in TBS-Tw for more than 2 h. After washed with TBS-Tw, the membrane was incubated with peroxidase-conjugated secondary antibodies. After another round of washes, the membrane was incubated with a chemiluminescence substrate for peroxidase (WBKLS0500, Merck). Luminescent images were acquired by using an image analyzer (Amersham Imager 680 [version 2.0.0], Cytiva).

**Statistics and reproducibility**. All data sets were handled with the Excel software (Excel for Mac version 16.x, Microsoft). Statistical analyses and graph drawing were carried out with the Prism software (versions 7 and 9, GraphPad software). *P* values were assessed by unpaired, two-tailed *t*-test with Welch's correction. Statistic sources are provided in the Source Data file. The sample size is indicated in the figure legends. All experiments were reproduced at least three times. The representative results, out of these independent experiments, are shown.

**Reporting summary**. Further information on research design is available in the Nature Research Reporting Summary linked to this article.

## Data availability

Source data for all statistical analyses (Figs. 1b, d, e, g, h, 2d, e, g, 3b, d, 4b, 5d, f, and Supplementary Figs. 1b, 2b, c) and uncropped image of gel and blot (Figs. 2b, 3a, c, and Supplementary Figs. 2a) are provided in the Source Data file. All other data that support the finding of this study are available from the corresponding author on reasonable request. Source data are provided with this paper.

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

## Acknowledgements

We thank Y. Azuma and K. Ohsumi for reagents; T. Totsuka, M. Shintomi, and M. Ohsugi for their help with the preparation of mouse sperm nuclei; M. Shima for her help with preparation of recombinant proteins; and members of the Hirano laboratory for discussions and critical reading of the manuscript. This work was supported by Grant-in-Aid for Scientific Research, KAKENHI (grants 18H02381 and 19H05755 [to K.S.], and 15H05971, 18H05276 and 20H05938 [to T.H.]).

## Author contributions

K.S. prepared the materials and performed the experiments; K.S. and T.H. designed and analyzed the experiments, and wrote the manuscript.

## Competing interests

The authors declare no competing interests.
