## [Peer Review File · Nature Communications]

REVIEWER COMMENTS

Reviewer #1 (Remarks to the Author):

This manuscript by Shintomi and Hirano presents an interesting set of observations on the process of mitotic chromosome assembly. Previously, the same authors had demonstrated that chromosome assembly can be achieved using a defined set of purified proteins, however these reconstituted chromosomes were not identical to those observed when egg-extracts were used for assemblies. This drove the authors to fine-tune the conditions for their reconstitutions. They report specific concentrations of MgCl₂ and KCl in which the reconstituted chromosome more closely resemble chromosomes formed in the extracts. They show that these conditions are likely to be important for Top2 function and then further explore the contribution of the CTD domain of Top2 to the assembly process. In doing so, they make several interesting observations, some of which might have very important implications for our current understanding of the chromosome formation process. They report that Top2 has two distinct functions during the formation of chromosomes, first the individualization of chromatids, and second a function in promoting the thickening of individualised chromatids. They present data suggesting that these two functions are driven by decatenation between chromosomes (individualisation) and catenation within the same chromatid (thickening).

Finally, the authors extend their analysis of Top2's contribution to chromosome assembly to a situation where nucleosome assembly is also impaired. They show that under such conditions the presence of Top2 is important to prevent the formation of an aberrant organisation, that they call sparklers, which is formed due to the inappropriate presence/binding of linker histones. They show that these sparkler structures can be fully reversed by addition of Top2, and present some evidence suggesting that the CTD domain plays a role in preventing sparkler formation.

Overall, this study is well executed and contains important data that will be of great interest to scientists in the field of chromosome biology. For these reasons, this study is appropriate for publication. I have a few minor points for the authors to consider.

The main criticism I have is that although the argument that intra-chromatid thickening might occur through catenation makes sense, (if one considers that the environment of the chromosome at that stage would be very crowded and that indeed extruded loops are likely to provide substrate for Top2 to concatenate), it is difficult to prove this experimentally. I do feel that the topology assays on plasmids might not be fully reflective of what happens on the chromosome situation because it is a very simplified situation with only DNA and Top2. I am fully aware of the difficulty of gathering direct evidence for catenation activity in the chromosome assembly experiments, but I think it is important to state this limitation, as well as consider other possibilities (besides catenation), this could easily be addressed in the discussion.

Also, it would be nice for the authors to clarify whether they believe the second role of Top2 (i.e. intra-chromatid concatenation) to be unidirectional or bidirectional. This has implications for future studies because a unidirectional role would most likely require stabilisation of the intra-chromatid catenanes. Alternatively in a bidirectional function, the enlargement of the adjacent extruded loops would cause an increase in the number of catenations present between them at any given time as a function of their size (i.e. thus leading to stabilisation of the structures despite the bidirectionality of Top2). I suggest the authors expand some of these points in the discussion.

The argument that CTD has no effect in decatenation (individualisation) seems to be true for the specific salt conditions used in the reconstitution (looking at the decatenation assays on plasmids) because when salt conditions are increased, full length Top2 continues to decatenate but CTD mutant stops, which shows that at least under such higher salt conditions CTD mutant is also defective in decatenation? This highlights the limitation of directly extrapolating results in the plasmid assay with the observations on the chromosome assembly.

It would be quite nice to show a temporal separation of the 1st and 2nd role of Top2. Perhaps the authors could consider an experiment where they generate individualised (but thin) chromatids driven by the CTD mutant Top2, and then add full length Top2 to drive the second function of thickening.

Reviewer #2 (Remarks to the Author):

In this work, Shintomi and Hirano optimized their mitotic chromatid reconstitution buffer to further dissect mitotic functions of topo IIa. Next, they conducted reconstitution assays, enzymological assays and cell-free assays using FL topoIIa and Δ CTD topoIIa. The experimental results lead the authors to conclude: first, that CTD dependent intra-chromatid catenation underlies chromatid thickening; second, that the CTD competes with a linker histone B4 to prevent abnormal configurations (sparklers) during chromatid reconstitution.

The manuscript is clearly written and concise. The figures are well presented and the methods properly detailed.

The main experimental observations regarding the failure of chromatid thickening and the formation of sparklers are unambiguous and very interesting. I find however that the conclusions of the study assuming a specific role of CDS in regulating these structural transitions are a bit precipitate. Other plausible interpretations of the results (see below) must be ruled out. If so, these experiments deserve publication in one or even two independent papers since the mechanistic aspects of chromatid thickening and sparkler formation are probably distinct and reflect functionally separated processes.

Major concerns:

1) Failure of chromatid thickening using Δ CTD topo could be consequence to a rapid decay of the catalytic activity of this enzyme relative to the FL topo. I observe that purification yield of FL is higher (>10 fold) than that of Δ CTD. The amount of enzyme and incubation periods used in vitro decatenation-catenation assays indicate that the specific activity (% of catalytically active topoisomerase) of both enzymes is low. This is probably exacerbated by the presence of tags in the N- and C-termini. So I wonder whether the activity of these enzymes might decay quite rapidly when exposed to complex environments (reconstitution assays) for long incubation periods (150 min). Therefore, a crucial control is to demonstrate that the catalytic activity of FL and Δ CTD remains comparable until the final stages of the reconstitution assays. To this end, authors could add catenated DNA during late time points of the reconstitution processes and test whether FL and Δ CTD topoisomerases are still able to decatenate the input DNA. In addition, authors could show via western blots that the enzymes are not differently degraded during these long incubation times.

2) Differential decay of the FL and Δ CTD topo activities could explain also why both the formation of sparklers and the resolution of pre-assembled sparklers are rescued by FL and only partially rescued by Δ CTD.

Therefore, these reconstitution experiments would require similar controls that show comparable activity and integrity of both enzymes.

3) The authors specify that... "Because the conditions that support efficient DNA decatenation were comparable to those required for chromatid individualization in the reconstitution assay, it is reasonable to assume that topo IIa-catalyzed decatenation facilitates the chromatid individualization process"... They next state that.... "the conditions required for DNA catenation closely matched those for chromatid thickening in the reconstitution assay. We therefore hypothesized that CTD mediated binding of topo IIa to chromatid axes might contribute to chromatid thickening by increasing the chance to generate intra-chromatid catenanes"....

I believe that the above extrapolations of the in vitro decatenation-catenation results to assume the topological changes that chromatids undergo during reconstitution assays are also precipitated and should be toned down. Decatenation-catenation equilibria by topo II are very sensitive to reaction conditions, and the local concentrations of protein and DNA. Such extrapolation would be more convincing if authors were able to conduct the in vitro decatenation-catenation reactions of plasmid DNA in presence of the six complexes that support chromatid reconstitution, instead of the lonely topo II enzymes. I think this would be a viable and enlightening experiment.

Some aspects and interpretations of these in vitro experiments need also to be clarified. For instance:

4) Although Δ CTD failed to catenate DNA, it produced knotted DNAs at levels comparable to FL (fig 3d).

Since intra-chromatid catenanes are equivalent to knots, then the hypothesis that chromatid thickening is due to CTD mediated formation of knots does not sustain.

5) Regarding the above, in the blow-up of fig 2c, it seems that the chromatid gets thick because it coils on itself, apparently forming a superhelix. If this is the case, authors could discuss how such coiling would be driven by intra-chromatid catenanes or, alternatively, by proteinic reorganization of its axis (condensin?).

6) Authors use only AMP-PNP (instead of ATP) in the DNA catenation/knotting assay. Why not incubate with ATP?. This would reveal the true steady-state products of the FL and Δ CTD topo activities, and that would be more extrapolatable to interpret the reconstitution assays.

Minor comment

If the authors conclusions are corroborated, the title of the paper should be more informative by stating the regulatory role of CTDs in chromatid condensation decondensation.

Reviewer #3 (Remarks to the Author):

In the submitted manuscript "Guiding topoisomerase II to crowded environments created during chromosome assembly", K. Shintomi and T. Hirano present their in-depth analysis of the type II topoisomerase topo II α and its function in mitotic chromosome assembly. The work focuses on the function of the C-terminal domain (CTD) of topo II α and reveals a central function for this domain in facilitating chromatid thickening. Topo II α is known to play a fundamental, but still not fully understood, function in mitotic chromosome assembly and segregation. By revealing a function for topo II α outside its well-known role in decatenation of sister chromatids, the study by Shintomi & Hirano advances the field. By focusing on a central process in chromosome dynamics and revealing a new function for topo II α , the presented investigation becomes relevant for researchers within the field, as well as non-specialists.

With the aim to perform fine-tuned analysis of topo II α function, the authors first improve their already established mitotic chromatid reconstitution assay and identify buffer conditions that allow chromatid thickening and individualization more similar to what has been observed in *X. laevis* cell free extracts. Thereafter, recombinant *X. laevis* full length topo II α (topo II α -FL) and CTD-deleted (topo II α - Δ CTD) versions are analyzed for their ability to individualize and thicken chromatids in the reconstitution assay. This reveals that both forms are proficient for individualization in optimal buffer conditions, even if the levels of topo II α - Δ CTD on chromosomes is reduced as compared to topo II α -FL. At increased KCl, topo II α - Δ CTD fails to bind and individualize chromosomes. Chromatid thickening is also more pronounced in reactions containing topo II α -FL as compared to topo II α - Δ CTD. Further support for a role of CTD in chromatid thickening is obtained using *Xenopus* egg extracts, as well as *Mus musculus* sperm nuclei extracts, depleted of topo II α . When reconstituting these extracts with topo II α -FL individualization and thickening of chromatids is observed, while topo II α - Δ CTD supports individualization only, and is substantially less accumulated on chromatids.

To further understand the above presented results the authors also perform in vitro decatenation and catenation assays. These show that topo II α - Δ CTD is fully proficient in decatenation in optimal buffer conditions, but not at higher KCl concentration. They also reveal that topo II α - Δ CTD is deficient in catenation under both conditions. Based on this, the authors propose that the results obtained in the mitotic chromatid reconstitution assay reflect that CTD-dependent inter-chromatid catenation contributes to the thickening process. Supporting the presence of stable catenations (and arguing against a structural function for topo II α which has been suggested earlier), the authors move on to show that the structure of chromatids reconstituted with topo II α -FL remains largely unchanged after removal of the topoisomerase using high salt wash. If instead the

reconstitution is performed with topo II α - Δ CTD, the same treatment caused more dramatic alterations in chromatid structure, in line with deficient catenation.

Based on the above summarized results, the authors propose that topo II α , in addition to chromatid de-catenation, also promotes chromatid compaction (the observed thickening) during mitosis. This function is suggested to demand topo II α -dependent intra-chromatid catenation, which in turn depends on CTD-dependent enrichment of topo II α to chromatid axes. These assumptions have good support from the presented investigations, and by earlier analysis of chromatid stiffness (J Cell Biol 188, 653-663 (2010)).

In a last series of experiments the authors co-deplete topo II α and the histone chaperone Asf1 from mitotic extracts and find that highly compacted chromosome structures that are named "sparklers" are formed. These are expected to be nucleosome-free structures, and, accordingly, are shown to lack histone H3. However, "sparklers" contain the linker histone B4, and display specific distribution patterns of condensins I and II. The hyper-compaction and accumulation of B4 can be suppressed and resolved by re-addition of topo II α -FL, while topo II α - Δ CTD only have marginal effect on the "sparklers". The authors suggest that this reflects how topo II α competes with B4 for chromosome-association in a CTD-dependent manner, and thereafter aids in chromatid disentanglement. Even though plausible, additional explanations, such as topo II α -dependent resolution of a structure which is needed for the aberrant accumulation of B4, can be envisaged.

In general, the first part of the manuscript describing the link between CTD-dependent catenation and chromatid thickening is interesting and well executed, and the results advance the field. The quality of the analysis of "sparklers" is also high, but the logic behind this part of the study, and the potential relevance of the obtained results for the understanding of topo II α function *in vivo*, is less clear. The assumption that it reflects CTD-dependent recruitment of topo II α to "crowded environments" is logical, but the experimental support for this is not that strong.

In conclusion, the presented study increases the understanding of the function of topo II α and its CTD domain in the assembly of mitotic chromosomes. The investigation is generally well performed, and the conclusions drawn are reasonable, but the following points need to be addressed.

- The evidence for CTD-dependent recruitment of topo II α to "crowded environment" is not very strong and alternative models should be considered. Correspondingly, the title of the manuscript also needs to be changed.
- It would be interesting to further explore the author's proposal that topo II α -dependent catenation (and thickening) is connected to condensin function. The distinct patterns of distribution of condensins in the "sparklers" is also indicative of an active role in their formation. If possible, this could be experimentally addressed. If not, potential underlying reasons for the distinct binding patterns could be further discussed.
- The logic behind the analysis of nucleosome-free chromosomes would benefit from being more clearly explained. How (if at all) does the analysis of nucleosome-free chromatids relate to mitotic chromatid organization *in vivo*?
- Figure 2e shows "Profiles of normalized signal intensities of DAPI along lines drawn perpendicular to chromatid axes were analyzed. The mean \pm s.d. is shown (n=15 lines from 5 chromatids)". Since single chromatids appear to vary in density and width in both topo II α -FL and topo II α - Δ CTD conditions it is important to know the selection criteria for the 15 positions.
- The authors "...envision that topo II α catenates neighboring DNA loops created by condensins, thereby relieving topological stress and facilitating further loop extrusion and chromatid thickening". What type of topological stress? This could be more clearly defined.

Camilla Björkegren

Authors' reply

Shintomi and Hirano (NCOMMS-20-45316-T)

General comments to all reviewers

- (1) To make the title of the article more specific and informative, we have revised it to “Guiding functions of the C-terminal domain of topoisomerase II α advance mitotic chromosome assembly”.
- (2) We have refined the terminology for the linker histone present in *Xenopus* egg extracts. In the original manuscript, we called it “B4”, one of the conventional names used for the embryonic linker histone in *Xenopus laevis*. In the revised manuscript, we have decided to use the name “H1.8” according to the phylogeny-base nomenclature proposed by Talbert et al. (2012) [*Epigenetics Chromatin*, 5:7]. This change would benefit the broad readership of the journal.
- (3) We have rephased the word “intra-chromatid catenanes” with “intra-chromatid entanglements” because the latter word is more precise and is consistent with the literature (also see Reply to Reviewer #2 Comment 4).

Reviewer #1

(Comment 1)

The main criticism I have is that although the argument that intra-chromatid thickening might occur through catenation makes sense, (if one considers that the environment of the chromosome at that stage would be very crowded and that indeed extruded loops are likely to provide substrate for Top2 to concatenate), it is difficult to prove this experimentally. I do feel that the topology assays on plasmids might not be fully reflective of what happens on the chromosome situation because it is a very simplified situation with only DNA and Top2. I am fully aware of the difficulty of gathering direct evidence for catenation activity in the chromosome assembly experiments, but I think is important to state this limitation, as well as consider other possibilities (besides catenation), this could easily be addressed in the discussion.

(Reply)

We agree with this reviewer's comment. As the reviewer pointed out, the enzymological assays using simple DNA substrates might not fully recapitulate what happens in the context of large-scale chromosome assembly. One of the important directions in the future is to address how topo II α functions together with condensins. To this end, structural and functional assays using megabase-sized DNA would help fill the gap between the existing assays. We think, however, that establishment of such experimental setups is very challenging at this moment and is beyond

the scope of the current study. That said, to mention the limitation of our current experiments, we have placed the following sentence in Discussion (page 9).

Our current results do not exclude the possibility that the CTD might contribute to chromatid thickening through a mechanism(s) other than intra-chromatid entanglement.

(Comment 2)

Also, it would be nice for the authors to clarify whether they believe the second role of Top2 (i.e. intra-chromatid concatenation) to be unidirectional or bidirectional. This has implications for future studies because a unidirectional role would most likely require stabilisation of the intra-chromatid catenanes. Alternatively in a bidirectional function, the enlargement of the adjacent extruded loops would cause an increase in the number of catenations present between them at any given time as a function of their size (i.e. thus leading to stabilisation of the structures despite the bidirectionality of Top2). I suggest the authors expand some of these points in the discussion.

(Reply)

We appreciate this thoughtful comment. We speculate that the second role of topo II α would be bidirectional: topo II α not only introduces intra-chromatid entanglements but also resolves them in assembled chromosomes because topo II α is unable to recognize the direction of its reactions under such conditions. In other words, our results are simply reflective of an equilibrium of bidirectional reactions under a given condition. To make this point clearer, we have added the following sentence (page 9) in the revised manuscript:

It should also be noted that topo II α 's action under this condition would be bidirectional, keeping an equilibrium between entanglement and disentanglement.

(Comment 3)

The argument that CTD has no effect in decatenation (individualisation) seems to be true for the specific salt conditions used in the reconstitution (looking at the decatenation assays on plasmids) because when salt conditions are increased, full length Top2 continues to decatenate but CTD mutant stops, which shows that at least under such higher salt conditions CTD mutant is also defective in decatenation? This highlights the limitation of directly extrapolating results in the plasmid assay with the observations on the chromosome assembly.

(Reply)

We admit that our explanation for this point in the original manuscript was incomplete. It should be emphasized that, under the high salt condition (i.e., 150 mM KCl), topo II α - Δ CTD is defective not only in decatenating kinetoplast DNA in the decatenation assay (Fig. 3a, b) but also in producing fibrous chromatin structures in the chromatid reconstitution assay (Fig. 2f). The resultant structure produced in the latter assay resembles a banana-shaped structure observed in a low-salt reaction containing no topo II α (see Supplementary Fig. 1a). Thus, the requirements for kinetoplast DNA decatenation (the CTD and buffer conditions) closely parallel those for chromatid individualization, making it reasonable to speculate that the two reactions observed in the two different assays are in fact supported by the same mechanism of action of topo II α . To make this argument clearer, we have placed the following phrase in the revised manuscript (page 5):

When the same set of assays was repeated [...], but topo II α - Δ CTD failed to do so, leaving banana-shaped structures that resemble those produced in a reaction containing no topo II α at 80 mM KCl (Supplementary Fig. 1a). Note that both topo II α -FL and topo II α - Δ CTD were barely detectable on chromatin at 150 mM KCl (Fig. 2f, g; 150 mM KCl).

(Comment 4)

It would be quite nice to show a temporal separation of the 1st and 2nd role of Top2. Perhaps the authors could consider an experiment where they generate individualised (but thin) chromatids driven by the CTD mutant Top2, and then add full length Top2 to drive the second function of thickening.

(Reply)

We appreciate this constructive comment. The suggested experiment makes sense if chromatid individualization and thickening occur in a completely ordered fashion. Although conceptually separable, we think that under the standard condition the two processes are mechanistically coupled and proceed simultaneously. We therefore believe that the suggested experiment, although interesting, would not provide additional insights into the current manuscript.

Reviewer #2

(Comment 1)

Failure of chromatid thickening using Δ CTD topo could be consequence to a rapid decay of the catalytic activity of this enzyme relative to the FL topo. I observe that purification yield of FL is higher (>10 fold) than that of Δ CTD. The amount of enzyme and incubation periods used in

in vitro decatenation-catenation assays indicate that the specific activity (% of catalytically active topoisomerase) of both enzymes is low. This is probably exacerbated by the presence of tags in the N- and C-termini. So I wonder whether the activity of these enzymes might decay quite rapidly when exposed to complex environments (reconstitution assays) for long incubation periods (150 min). Therefore, a crucial control is to demonstrate that the catalytic activity of FL and Δ CTD remains comparable until the final stages of the reconstitution assays. To this end, authors could add catenated DNA during late time points of the reconstitution processes and test whether FL and Δ CTD topos are still able to decatenate the input DNA. In addition, authors could show via western blots that the enzymes are not differently degraded during these long incubation times.

(Reply)

We performed the experiment this reviewer had suggested. As depicted in the schematic diagram shown on the right (*upper*), a set of experiments were designed to test (1) whether the polypeptides of topo II α -FL and topo II α - Δ CTD remain intact after 150-min incubation in the reconstitution mixture, and (2) whether their decatenation activities remain intact after the same period of incubation. To this end, chromatid reconstitution mixtures

containing either topo II α -FL or topo II α - Δ CTD were divided into two, and the one was supplemented with catenated DNA (kinetoplast DNA) at 0 min (before chromatid assembly) and the other was supplemented at 150 min (after chromatid assembly). To test (1), aliquots were taken at 0 and 150 min and the amounts of topo II α in the assembly mixtures were analyzed by immunoblotting. To test (2), aliquots were taken 0, 15 and 60 min after each timing of adding catenated DNA, and subjected to the standard decatenation assay. Our results confirmed that both the amounts (*lower left*) and enzymatic activities (*lower right*) of topo II α -FL and topo II α - Δ CTD did not change during the 150-min incubation, thus eliminating the reviewer's concern that topo II α - Δ CTD may be less stable than topo II α -FL in the chromatid reconstitution mixtures.

As the reviewer pointed out, the purification yield of topo II α - Δ CTD was much lower

than that of topo II α -FL. At this moment, we do not know the exact reason for this. Given the clear results shown above, however, we are confident that the two proteins have the same level of specific activities after purification and retain them even after 150-min incubation in the chromatid assembly mixtures.

(Comment 2)

Differential decay of the FL and Δ CTD topo activities could explain also why both the formation of sparklers and the resolution of pre-assembled sparklers are rescued by FL and only partially rescued by Δ CTD. Therefore, these reconstitution experiments would require similar controls that show comparable activity and integrity of both enzymes.

(Reply)

We repeated the same set of experiments as above using a mitotic egg extract (M-HSS) depleted of topo II α and Asf1, in which sparklers were formed (*upper*). The results of an immunoblot analysis (*lower left*) and a decatenation assay (*lower right*) clearly demonstrate that topo II α -FL and topo II α - Δ CTD are stable and retain their decatenation activities even after 150-min incubation in the M-HSS.

(Comment 3)

The authors specify that... “Because the conditions that support efficient DNA decatenation were comparable to those required for chromatid individualization in the reconstitution assay, it is reasonable to assume that topo II α -catalyzed decatenation facilitates the chromatid individualization process”.... They next state that.... “the conditions required for DNA catenation closely matched those for chromatid thickening in the reconstitution assay. We therefore hypothesized that CTD mediated binding of topo II α to chromatid axes might contribute to chromatid thickening by increasing the chance to generate intra-chromatid catenanes”.

I believe that the above extrapolations of the *in vitro* decatenation-catenation results to

assume the topological changes that chromatids undergo during reconstitution assays are also precipitated and should be toned down. Decatenation-catenation equilibria by topo II are very sensitive to reaction conditions, and the local concentrations of protein and DNA. Such extrapolation would be more convincing if authors were able to conduct the *in vitro* decatenation-catenation reactions of plasmid DNA in presence of the six complexes that support chromatid reconstitution, instead of the lonely topo II enzymes. I think this would be a viable and enlightening experiment.

(Reply)

As the reviewer pointed out, we admit that the decatenation and catenation assays using simple DNA substrates might not fully recapitulate what happens on chromosomal DNA in the chromatid reconstitution assay. We also think it important to understand how topo II α changes the topology of nucleosomal DNA in the presence of condensin I and histone chaperones. However, we want to remind the reviewer that the protein mixture used in the current reconstitution assay lacks histone H3-H4 and its chaperones, thereby being unable to assemble nucleosome on the circular DNA template (Note that the mixture can assemble nucleosomes on the substrate of *Xenopus* sperm nuclei because they contain an adequate amount of H3-H4 [Shintomi et al, 2015, *Nat Cell Biol*]). For this reason, the experiment the reviewer suggested is not feasible at this moment and the establishment of such an experimental setup is beyond the scope of the current study. That said, to mention the limitation of our current experiments, we have placed the following sentence in Discussion (page 9).

Our current results do not exclude the possibility that the CTD might contribute to chromatid thickening through a mechanism(s) other than intra-chromatid entanglement.

(Comment 4)

Although Δ CTD failed to catenate DNA, it produced knotted DNAs at levels comparable to FL (fig 3d). Since intra-chromatid catenanes are equivalent to knots, then the hypothesis that chromatid thickening is due to CTD mediated formation of knots does not sustain.

(Reply)

First of all, let us explain our wording for the products in the DNA catenation/knotting assay (Fig. 3c). As the reviewer correctly pointed out, “intra-molecular” strand-passage results in the formation of knotted DNAs in this setup. On the other hand, “inter-molecular” strand-passage results in the formation of catenated DNA networks. To readily discriminate these two DNA species produced through distinct modes of catalytic actions of topo II α , we used the words of

“knotted” and “catenated” along with careful descriptions about their differences in the original text (page 5). “At 80 mM KCl, topo II α -FL generated two types of DNA products [...]: fast-migrating knotted DNAs made from single DNA molecules and slowly-migrating catenated DNAs made from multiple DNA molecules (Fig. 3c, d)”. We think that keeping such a wording is beneficial to readers because it has been used since earlier works in the field (e.g., Hsieh, 1983, *J Biol Chem* [ref. 21]; Hirose et al, 1988, *J Biol Chem* [ref. 25]). It is therefore reasonable to conclude that the CTD is required for DNA catenation in this particular experimental setup.

Then the question is how the terminology used for the DNA catenation/knotting assay should be related to those for the chromatid reconstitution assay. We found that requirements for chromatid thickening in the reconstitution assay (Fig. 2) are very similar to those for (inter-molecular) catenation in the DNA catenation/knotting assay (Fig. 3c, d), enabling us to hypothesize that CTD-mediated chromatin binding of topo II α increases the chance to catenate *spatially distant* DNA segments present in different loops within a single chromatid. In this sense, the word “intra-chromatid catenanes” could be better described as “inter-loop catenanes present in the same chromosomal DNA”. Thus, according to our terminology, knots and catenanes represent intra-loop entanglements and inter-loop catenanes, respectively, and the reviewer’s argument that “intra-chromatid catenanes are equivalent to knots” is not appropriate to describe our results and interpretations. That said, we now recognize that the use of the word “catenanes” in the context of chromosomal DNA is potentially confusing. In fact, a previous paper by Kawamura et al (2010, *J Cell Biol*) referred to the corresponding structures as “intra-chromatid entanglements”. To make these points clearer, we have rephrased the corresponding sentence as follows (page 6):

We therefore hypothesized that CTD-mediated binding of topo II α to chromatid axes might contribute to chromatid thickening by increasing the chance to generate entanglements between different chromatin loops within the same chromosomal DNA (hereafter, referred to as “intra-chromatid” entanglements).

(Comment 5)

Regarding the above, in the blow-up of fig 2c, it seems that the chromatid gets thick because it coils on itself, apparently forming a superhelix. if this is the case, authors could discuss how such coiling would be driven by of intra-chromatid catenanes or, alternatively, by proteinic reorganization of its axis (condensin?).

(Reply)

We appreciate this comment. As this reviewer pointed out, one potential mechanism behind the

thickening process is further coiling of a chromatid fiber. In fact, we have already suggested this possibility in the Discussion section (page 9) by citing a recent Hi-C study that had proposed helical folding of mitotic chromosomes (Gibcus et al 2018, *Science* [ref 15]): “It is also possible that intra-chromatid entanglements are generated between DNA loops separated by a great genomic distance, for instance, when they are brought closer through a mechanism of helical winding of chromatid axes.”

Elucidating the molecular mechanism of chromatid thickening is an exciting question to be addressed in the future, but it is beyond the scope of the current study.

(Comment 6)

Authors use only AMP-PNP (instead of ATP) in the DNA catenation/knotting assay. Why not incubate with ATP? This would reveal the true steady-state products of the FL and Δ CTD topo activities, and that would be more extrapolatable to interpret the reconstitution assays.

(Reply)

First of all, let us explain the historical background of DNA catenation/knotting assays. We followed the protocol reported by Roca et al. (1993, *J Biol Chem*), in which AMP-PNP was used to allow only a single round of strand passage reactions mediated by budding yeast topo II. In our original manuscript, we wanted to reproduce their results by using recombinant *Xenopus* topo II α , and this was the reason why we used AMP-PNP instead of ATP in this particular experiment. We are aware, however, that another paper had reported that *Drosophila* topo II can support DNA catenation/knotting in the presence of ATP (Hsieh, 1983, *J Biol Chem*). For

this reason and according to the reviewer’s suggestion, we now have repeated our DNA catenation/knotting assay in the presence of ATP. As shown here, ATP can facilitate catenation reactions dependently of the CTD in our experimental setup although the effect of ATP was less efficient than that of AMP-PNP. As the reviewer pointed out, the ATP-driven reaction would better reflect the steady-state products of topo II α ’s catalytic reactions. However, what we want to demonstrate in the current manuscript is that topo II-mediated catenation required the CTD. We therefore consider that presenting the results using AMP-PNP are sufficient.

(Comment 7)

If the authors conclusions are corroborated, the title of the paper should be more informative by stating the regulatory role of CTDs in chromatid condensation/decondensation.

(Reply)

According to the reviewer's suggestion, we have decided to include the specific word "the C-terminal domain" into the title. The new title of the revised manuscript is "Guiding functions of the C-terminal domain of topoisomerase II α advance mitotic chromosome assembly".

Reviewer #3

(Comment 1)

The evidence for CTD-dependent recruitment of topo II α to "crowded environment" is not very strong and alternative models should be considered. Correspondingly, the title of the manuscript also needs to be changed.

(Reply)

According to this reviewer's comment, we have decided to drop the word "crowded environments" from the title, and revised it to "Guiding functions of the C-terminal domain of topoisomerase II α advance mitotic chromosome assembly". We believe that the new title is more specific and objective than the original one.

(Comment 2)

It would be interesting to further explore the author's proposal that topo II α -dependent catenation (and thickening) is connected to condensin function. The distinct patterns of distribution of condensins in the "sparklers" is also indicative of an active role in their formation. If possible, this could be experimentally addressed. If not, potential underlying reasons for the distinct binding patterns could be further discussed.

(Reply)

As this reviewer correctly pointed out, how the topo II α -dependent catenation proposed in the current manuscript is mechanistically connected to condensins' function is a very interesting question. We think, however, that experimentally addressing this question is clearly beyond the scope of the current study. We are also curious to know whether condensins I and II have active roles in sparkler formation. One of the most direct tests for this question would be to deplete either or both of condensins from egg extracts along with Asf1 and topo II α . Unfortunately, triple- or even quadruple-depletion (namely, Asf1 and topo II α plus condensin I and/or II) using

currently available antibodies is technically challenging, and we have not been able to establish such a protocol so far. To emphasize that testing these points is one of the important directions in the future, we have placed the following sentence in Discussion (page 10):

Further studies will be required to understand the molecular mechanism behind sparkler formation and possible involvements of other protein components including condensins I and II.

(Comment 3)

The logic behind the analysis of nucleosome-free chromosomes would benefit from being more clearly explained. How (if at all) does the analysis of nucleosome-free chromatids relate to mitotic chromatid organization *in vivo*?

(Reply)

Although we attempted, in the original manuscript (page 6), to explain our logic behind the analysis of topo II α 's functions in nucleosome-free chromatid assembly, it may not have been sufficient. We hope that the following explanation will help the reviewer understand our thought on this issue.

One of the most important conclusions in the chromatid reconstitution assays (Figs. 1 and 2) is that CTD-mediated chromatin binding of topo II α is essential for chromatid thickening. This finding was also confirmed in *Xenopus* egg cell-free extracts, through simple depletion and add-back assays (Supplementary fig. 2). At that point, we wondered whether any other experimental conditions might help to further illuminate the importance of the CTD during chromatid assembly. In our previous study, we had noticed that topo II α localizes along the entire length of nucleosome-free chromatids which was assembled from mouse sperm nuclei incubated with an Asf1-depleted extract (Shintomi et al, 2017, Science). Taken all into consideration, we wished to know what would happen upon double depletion of Asf1 and topo II α , and whether defects caused under this condition could be rescued by adding back recombinant topo II α . In this line, we were first surprised to find that the double depletion caused production of an unprecedented structure, which we called sparkers (Fig. 4). But we were also able to demonstrate that topo II α -FL, but not topo II α - Δ CTD, can suppress the formation of sparklers (and resolve preformed sparklers), efficiently promoting nucleosome-free chromatids. It should be emphasized that the assembly of nucleosome-free chromatids (and sparklers) is possible only in the combination of mouse sperm nuclei and *Xenopus* egg cell-free extracts: it is not possible in the chromatid reconstitution assay using frog sperm nuclei. We reasoned that the unique results from the cell-free extracts significantly broaden our

understanding of mitotic functions of topo II α , and therefore decided to include them in the current manuscript.

(Comment 4)

Figure 2e shows “Profiles of normalized signal intensities of DAPI along lines drawn perpendicular to chromatid axes were analyzed. The mean +/- s.d. is shown (n=15 lines from 5 chromatids)”. Since single chromatids appear to vary in density and width in both topo II α -FL and topo II α - Δ CTD conditions it is important to know the selection criteria for the 15 positions.

(Reply)

Admittedly our description of this analysis was insufficient in the original text. As the reviewer pointed out, the width of chromatids vary to some extent even within individual chromatids. To avoid biased measurements, we drew a first line perpendicular to the chromatid axis at the widest region on a chromatid and then added two parallel lines on both sides of the first line at a 1- μ m distance. DAPI density along these three lines were scanned. The same procedure was repeated for four additional chromatids (n=15 from 5 chromatids). In the revised manuscript, we have provided the same explanation in the Methods section (page 15).

(Comment 5)

The authors “...envision that topo II α catenates neighboring DNA loops created by condensins, thereby relieving topological stress and facilitating further loop extrusion and chromatid thickening”. What type of topological stress? This could be more clearly defined.

(Reply)

We appreciate this comment. The meaning of “topological stress” was too vague to precisely describe chromatin dynamics in our model. We just wanted to consider the possibility that strand passage between neighboring DNA loops could decrease their steric hindrance. To clarify this point, we have revised the corresponding text as follows (page 9):

We envision that topo II α allows entanglement between neighboring DNA loops created by condensins, thereby decreasing steric hindrance among DNA segments and facilitating further loop extrusion and chromatid thickening (Fig. 6a).

REVIEWERS' COMMENTS

Reviewer #1 (Remarks to the Author):

The authors have addressed all my concerns in the response. I am supportive of publication.

Reviewer #2 (Remarks to the Author):

In the revised version of the manuscript, the authors have satisfactorily addressed most of my concerns and I am therefore happy to recommend its publication. The new title and the changes in the result and discussion sections make this a much clearer and convincing contribution.

Reviewer #3 (Remarks to the Author):

The authors have fully addressed the concerns raised regarding the first version of the manuscript. The new title: "Guiding functions of the C-terminal domain of topoisomerase IIa advance mitotic chromosome assembly" might be somewhat complicated for the non-specialist reader and could possibly be simplified. Otherwise, I have no further comments.